# DATA VALUATION WITHOUT TRAINING OF A MODEL

**Nohyun Ki**,* **Hoyong Choi**,* **Hye Won Chung**
School of Electrical Engineering
Korea Advanced Institute of Science and Technology (KAIST)
Daejeon, South Korea
{kinohyun, chy0707, hwchung}@kaist.ac.kr

## ABSTRACT

Many recent works on understanding deep learning try to quantify how much individual data instances influence the optimization and generalization of a model. Such attempts reveal characteristics and importance of individual instances, which may provide useful information in diagnosing and improving deep learning. However, most of the existing works on data valuation require actual training of a model, which often demands high-computational cost. In this paper, we provide a training-free data valuation score, called *complexity-gap score*, which is a data-centric score to quantify the influence of individual instances in generalization of two-layer overparameterized neural networks. The proposed score can quantify irregularity of the instances and measure how much each data instance contributes in the total movement of the network parameters during training. We theoretically analyze and empirically demonstrate the effectiveness of the complexity-gap score in finding 'irregular or mislabeled' data instances, and also provide applications of the score in analyzing datasets and diagnosing training dynamics. Our code is publicly available at https://github.com/JJchy/CG_score.

## 1 INTRODUCTION

Creation of large datasets has driven development of deep learning in diverse applications including computer vision (Krizhevsky et al., 2012; Dosovitskiy et al., 2021), natural language processing (Vaswani et al., 2017; Brown et al., 2020) and reinforcement learning (Mnih et al., 2015; Silver et al., 2016). To utilize the dataset in a more efficient and effective manner, some recent works have attempted to understand the role of *individual* data instances in training and generalization of neural networks. In (Ghorbani & Zou, 2019), a metric to quantify the contribution of each training instance in achieving a high test accuracy was analyzed under the assumption that not only the training data but also the test data is available. Jiang et al. (2021) defined a score to identify irregular examples that need to be memorized during training, in order for the model to accurately classify the example.

All these previous methods for data valuation require *actual training* of a model to quantify the role of individual instances at the model. Thus, the valuation itself often requires high-computational cost, which may contradict some motivations of data valuation. For example, in (Ghorbani & Zou, 2019; Jiang et al., 2021), to examine the effect of individual data instances in training, one needs to train a model repeatedly while eliminating each instance or subsets of instances. In (Swayamdipta et al., 2020; Toneva et al., 2019), on the other hand, training dynamics–the behavior of a model on each instance throughout the training–is analyzed to categorize data instances. When the motivation for data valuation lies at finding a subset of data that can approximate the full-dataset training to save the computational cost for training, the previous valuation methods might not be suitable, since they already require the training with the full dataset before one can figure out 'important' instances.

In this paper, our main contribution is on defining a *training-free* data valuation score, which can be directly computed from data and can effectively quantify the impact of individual instances in optimization and generalization of neural networks. The proposed score, called *complexity-gap score*, measures the gap in *data complexity* where a certain data instance is removed from the full dataset. The data complexity measure was originally introduced in Arora et al. (2019) to quantify the complexity of the full dataset, which was used in bounding the generalization error of overparameterized

---

*Equal contribution.

two-layer neural networks trained by gradient descent. Different from that work, where the complexity of the full dataset was of main concern, our focus is on decomposing the effect of individual data instances in the training, and thus we newly introduce a complexity gap score (CG-score). We theoretically analyze and empirically demonstrate that the CG-score can quantify 'irregularity' of instances within each class, and thus can be used in identifying atypical examples, either due to the inherent irregularity of the instance or mislabeled classification. We also demonstrate that the proposed score has a close relation to 'learning difficulty' of the instances by analyzing the training dynamics of data instances. Our key contributions are as below:

- *Training-free data valuation:* Different from previous methods for data valuation, most of which leverage the information from training itself, we provide a training-free data valuation score, CG-score, which is the data-centric score to quantify the effect of individual data instances in optimization and generalization of neural networks.

- *Geometric interpretation:* We provide analysis that the CG-score can measure *irregularity* of each instance, i.e., it measures the average 'similarity' of an instance to the instances of the same class and the average 'dissimilarity' to the instances of different classes.

- *Effectiveness of the score*: We empirically demonstrate the effectiveness of the CG-score in data valuation. We show that pruning data instances with small CG-score does not significantly degrade the generalization capability of a model, e.g., for CIFAR-10 we can prune 40% of the data with less than 1% of drop in test accuracy. Our scoring method is especially useful in data pruning, since different from other scores, which require the training with the full dataset, our method does not require any training of a model.

- *Application of the score:* We provide potential applications of the CG-score in analyzing datasets and training dynamics. We analyze the histograms of the CG-score for various datasets to demonstrate that the CG-score can measure irregularity of the instances. We also demonstrate that the instances with higher CG-score are 'difficult' examples, which are learned slowly by the models, by comparing the loss and test accuracy curves and the evolution of Neural Tangent Kernel (NTK) submatrices of lowest/highest-scoring groups.

## 2 RELATED WORKS

Different from many existing works where the effect of datasets on model training is analyzed as a whole, some recent works have focused on understanding the impact of individual data instances. Ghorbani & Zou (2019); Kwon et al. (2021) defined 'Data Shapley', to evaluate the value of each data instance, by measuring the average gap in performances when an instance is held-out from any subsets of a given training data. Jiang et al. (2021) defined consistency score (C-score) of each instance by estimating the prediction accuracy of the instance attained by the model trained with the full dataset except the instance. Both Data Shapley and C-score require multiple trainings of a model to compute the scores. Another main branch uses the training dynamics to identify 'difficult' instances for classification, either due to irregularity or mislabeling, by measuring different forms of *confidence, stability or influence* in the decision of the networks throughout the training (Toneva et al., 2019; Swayamdipta et al., 2020; Pruthi et al., 2020). In (Baldock et al., 2021), the computational difficulty of an instance is defined as the number of hidden layers after which the networks' prediction coincides with the prediction at the output. With application of robust learning, some works quantify the difficulty of each instance by a 'margin' from the decision boundary (Zhang et al., 2021). CRAIG (Mirzasoleiman et al., 2020) finds valuable subsets of data as coresets that preserve the gradient of the total loss. All these previous methods are demonstrated to be effective in at least one or more applications of data valuation, including data pruning (Paul et al., 2021; Swayamdipta et al., 2020; Agarwal et al., 2022; Feldman & Zhang, 2020), importance-based weighted sampling (Chang et al., 2017; Koh & Liang, 2017), noise filtering (Li et al., 2020; Lee et al., 2019b; Kim et al., 2021), robust learning (Ren et al., 2018; Pleiss et al., 2020), out-of-distribution generalizations (Swayamdipta et al., 2020) or diverse sample generalization (Lee et al., 2021). However, all these methods require the training of a model with the full dataset (at least for a few optimization steps). Recently, some works try to evaluate different subsets of training data by utilizing generalization error bound, induced by the NTK theory of neural networks (Wu et al., 2022; Zhang & Zhang, 2022). However, these methods do not provide a computationally-efficient way to quantify the effect of individual data instances. Our data valuation method, on the other hand, is a data-centric method that can be efficiently calculated from data only without training of a model.

## 3   COMPLEXITY-GAP SCORE: DATA VALUATION WITHOUT TRAINING

In this section, we introduce a new data valuation score, called *complexity-gap score*, based on the analysis of overparameterized two-layer neural networks from Arora et al. (2019).

### 3.1   PRELIMINARIES: DATA COMPLEXITY MEASURE IN TWO-LAYER NEURAL NETWORKS

We first review the result from Arora et al. (2019), where a two-layer neural network trained by randomly initialized gradient descent is analyzed. Consider a two-layer ReLU activated neural network having $m$ neurons in the hidden layer, of which the output is $f_{\mathbf{W},\mathbf{a}}(\mathbf{x}) = \frac{1}{\sqrt{m}} \sum_{r=1}^{m} a_r \sigma(\mathbf{w}_r^\top \mathbf{x})$ where $\mathbf{x} \in \mathbb{R}^d$ is the input, $\mathbf{w}_1, \ldots, \mathbf{w}_m \in \mathbb{R}^d$ are weight vectors in the first layer, $a_1, \ldots, a_m \in \mathbb{R}$ are weights in the second layer, and $\sigma(x) = \max(0, x)$ is the ReLU activation function. Let $\mathbf{W} = (\mathbf{w}_1, \ldots, \mathbf{w}_m) \in \mathbb{R}^{d \times m}$ and $\mathbf{a} = (a_1, \ldots, a_m)^\top \in \mathbb{R}^m$. Assume that the network parameters are randomly initialized as $\mathbf{w}_r(0) \sim \mathcal{N}(0, \kappa^2 I_{d \times d})$ and $a_r \sim \mathrm{unif}(\{-1, 1\}), \forall r \in [m]$, where $\kappa \in (0, 1]$ is the size of random initialization. The second layer $\mathbf{a}$ is then fixed and only the first layer $\mathbf{W}$ is optimized through gradient descent (GD) to minimize the quadratic loss, $\Phi(\mathbf{W}) = \frac{1}{2} \sum_{i=1}^{n} (y_i - u_i)^2$ where $u_i = f_{\mathbf{W},\mathbf{a}}(\mathbf{x}_i)$ and $\{(\mathbf{x}_i, y_i)\}_{i=1}^{n}$ is the dataset drawn i.i.d. from an underlying distribution $\mathcal{D}$. The GD update rule can be written as $\mathbf{w}_r(k+1) - \mathbf{w}_r(k) = -\eta \frac{\partial \Phi(\mathbf{W}(k))}{\partial \mathbf{w}_r}$ where $\eta > 0$ is the learning rate. The output of the network for the input $\mathbf{x}_i$ at the $k$-th iteration is denoted by $u_i(k) = f_{\mathbf{W}(k),\mathbf{a}}(\mathbf{x}_i)$. For simplicity, it is assumed that $\|\mathbf{x}\|_2 = 1$ and $|y| \leq 1$.

The data complexity measure governing the training of the two-layer ReLU activated neural network is defined in terms of the following *Gram matrix* $\mathbf{H}^\infty \in \mathbb{R}^{n \times n}$ associated with ReLU activation:

$$\mathbf{H}^\infty_{ij} = \mathbb{E}_{\mathbf{w} \sim \mathcal{N}(0, I_{d \times d})} \left[ \mathbf{x}_i^\top \mathbf{x}_j \mathbb{1}\{\mathbf{w}^\top \mathbf{x}_i \geq 0, \mathbf{w}^\top \mathbf{x}_j \geq 0\} \right] = \frac{\mathbf{x}_i^\top \mathbf{x}_j (\pi - \arccos(\mathbf{x}_i^\top \mathbf{x}_j))}{2\pi}. \quad (1)$$

In Arora et al. (2019), a complexity measure of data was defined as $\mathbf{y}^\top (\mathbf{H}^\infty)^{-1} \mathbf{y}$ where $\mathbf{y} = (y_1, \ldots, y_n)$, and it was shown that this measure bounds the total movement of all neurons in $\mathbf{W}$ from their random initialization. Moreover, the data complexity measure bounds the generalization error by restricting the Rademacher complexity of the resulting function class. In the following, we write the eigen-decomposition of $\mathbf{H}^\infty$ as $\mathbf{H}^\infty = \sum_{i=1}^{n} \lambda_i \mathbf{v}_i \mathbf{v}_i^\top$ where $\lambda_i$'s are ordered such that $\lambda_1 \geq \lambda_2 \geq \ldots \lambda_n$. Further, assuming $\lambda_n \geq \lambda_0 > 0$, we can write $(\mathbf{H}^\infty)^{-1} = \sum_{i=1}^{n} (\lambda_i)^{-1} \mathbf{v}_i \mathbf{v}_i^\top$.

**Theorem 1** (Informal version of (Arora et al., 2019)). *Assume that $\lambda_{\min}(\mathbf{H}^\infty) = \lambda_n \geq \lambda_0 > 0$. For sufficiently large width $m$, sufficiently small learning rate $\eta > 0$ and sufficiently small random initialization $\kappa > 0$, with probability at least $1 - \delta$ over the random initialization, we have*

*a) Bound in loss :* $\|\mathbf{y} - \mathbf{u}(k)\|_2 = \sqrt{\sum_{i=1}^{n} (1 - \eta\lambda_i)^{2k} (\mathbf{v}_i^\top \mathbf{y})^2} + \textit{small constant,}$

*b) Bound in total movement of neurons:* $\|\mathbf{W}(k) - \mathbf{W}(0)\|_F \leq \sqrt{\mathbf{y}^\top (\mathbf{H}^\infty)^{-1} \mathbf{y}} + \textit{small constant,}$

*c) Bound in population loss:* $\mathbb{E}_{(\mathbf{x},y) \sim \mathcal{D}}[l(f_{\mathbf{W}(k),\mathbf{a}}(\mathbf{x}), y)] \leq \sqrt{\dfrac{\mathbf{y}^\top (\mathbf{H}^\infty)^{-1} \mathbf{y}}{n}} + O\left(\sqrt{\dfrac{\log \frac{n}{\lambda_0 \delta}}{n}}\right),$

*where a) and b) hold for all iteration $k \geq 0$ of GD, and c) holds for $k \geq \Omega\left(1/(\eta\lambda_0) \log(n/\delta)\right)$.*

This theorem shows that if the label vector $\mathbf{y}$ is aligned with top eigenvectors of $\mathbf{H}^\infty$, i.e., $(\mathbf{v}_i^\top \mathbf{y})$ is large for large $\lambda_i$, then the loss decreases quickly and the total movement of neurons as well as the generalization error is small. Thus, the data complexity measure $\mathbf{y}^\top (\mathbf{H}^\infty)^{-1} \mathbf{y}$ captures the complexity of data governing both the optimization and generalization of the overparameterized two-layer neural networks. However, this quantity captures the complexity of the whole data. To decompose the effect of individual instances, we newly define a *complexity-gap score*.

### 3.2   COMPLEXITY-GAP SCORE AND TRAINING DYNAMICS

We define the *complexity-gap score* (CG-score) of $(\mathbf{x}_i, y_i)$ as the difference between the data complexity measure when $(\mathbf{x}_i, y_i)$ is removed from a given dataset $\{(\mathbf{x}_i, y_i)\}_{i=1}^{n}$:

$$\mathrm{CG}(i) = \mathbf{y}^\top (\mathbf{H}^\infty)^{-1} \mathbf{y} - \mathbf{y}_{-i}^\top (\mathbf{H}^\infty_{-i})^{-1} \mathbf{y}_{-i} \quad (2)$$

where $\mathbf{y}_{-i}$ is the label vector except the $i$-th sample point and $\mathbf{H}_{-i}^{\infty}$ is the $(n-1) \times (n-1)$ matrix obtained by removing the $i$-th row and column of $\mathbf{H}^{\infty}$.

We first emphasize that the proposed score can be easily calculated from given data without the need of training neural networks, as opposed to other data valuation scores requiring either a trained neural network or statistics calculated from training dynamics. Yet, the proposed score captures two important properties on the training and generalization of data instance, implied by Theorem 1:

1. An instance $(\mathbf{x}_i, y_i)$ with a large CG-score is a 'difficult' example, in the sense that removing it from the dataset reduces the generalization error bound by a large amount, which implies that the dataset without $(\mathbf{x}_i, y_i)$ is much easier to be learned and generalized.

2. An instance $(\mathbf{x}_i, y_i)$ with a larger CG-score contributes more on the optimization and drives the total movement of neurons by a larger amount, measured by $\|\mathbf{W}(k) - \mathbf{W}(0)\|_F$.

We next discuss the computational complexity of calculating the CG-score. To calculate $\{CG(i)\}_{i=1}^n$, we need to take the inverse of matrices $\mathbf{H}^{\infty}$ and $\mathbf{H}_{-i}^{\infty}$ for all $i \in [n]$, which requires $O(n^4)$ complexity when we use general $O(n^3)$-complexity algorithm for the matrix inversion. By using Schur complement, however, we can reduce this complexity to $O(n^3)$. Without loss of generality, we can assume $i = n$. Denote $\mathbf{H}^{\infty}$ and $(\mathbf{H}^{\infty})^{-1}$ by

$$\mathbf{H}^{\infty} = \begin{pmatrix} \mathbf{H}_{n-1}^{\infty} & \mathbf{g}_i \\ \mathbf{g}_i^{\top} & c_i \end{pmatrix}, \quad (\mathbf{H}^{\infty})^{-1} = \begin{pmatrix} (\mathbf{H}^{\infty})_{n-1}^{-1} & \mathbf{h}_i \\ \mathbf{h}_i^{\top} & d_i \end{pmatrix}, \tag{3}$$

where $\mathbf{H}_{n-1}^{\infty}, (\mathbf{H}^{\infty})_{n-1}^{-1} \in \mathbb{R}^{(n-1) \times (n-1)}$, $\mathbf{g}_i, \mathbf{h}_i \in \mathbb{R}^{n-1}$ and $c_i, d_i \in \mathbb{R}$. From $\mathbf{H}^{\infty}(\mathbf{H}^{\infty})^{-1} = \mathbf{I}_n$, we have $\mathbf{g}_i^{\top}(\mathbf{H}^{\infty})_{n-1}^{-1} + c_i \mathbf{h}_i^{\top} = 0$, i.e., $\mathbf{h}_i^{\top} = -c_i^{-1} \mathbf{g}_i^{\top}(\mathbf{H}^{\infty})_{n-1}^{-1}$.

By Schur complement, $(\mathbf{H}_{-i}^{\infty})^{-1}$, which is equal to $(\mathbf{H}_{n-1}^{\infty})^{-1}$ for $i = n$, can be calculated as

$$(\mathbf{H}_{-i}^{\infty})^{-1} = (\mathbf{H}^{\infty})_{n-1}^{-1} - d_i^{-1} \mathbf{h}_i \mathbf{h}_i^{\top}. \tag{4}$$

Since we have

$$\mathbf{y}^{\top}(\mathbf{H}^{\infty})^{-1}\mathbf{y} = \mathbf{y}_{-i}^{\top}(\mathbf{H}^{\infty})_{n-1}^{-1}\mathbf{y}_{-i} + y_i \mathbf{h}_i^{\top}\mathbf{y}_{-i} + y_i \mathbf{y}_{-i}^{\top}\mathbf{h}_i + y_i^2 d_i,$$
$$\mathbf{y}_{-i}^{\top}(\mathbf{H}_{-i}^{\infty})^{-1}\mathbf{y}_{-i} = \mathbf{y}_{-i}^{\top}(\mathbf{H}^{\infty})_{n-1}^{-1}\mathbf{y}_{-i} - d_i^{-1}(\mathbf{y}_{-i}^{\top}\mathbf{h}_i)^2, \tag{5}$$

the CG-score, $CG(i)$ in equation 2, can be calculated by

$$CG(i) = d_i^{-1}(\mathbf{y}_{-i}^{\top}\mathbf{h}_i)^2 + 2y_i(\mathbf{y}_{-i}^{\top}\mathbf{h}_i) + y_i^2 d_i = \left((\mathbf{y}_{-i}^{\top}\mathbf{h}_i)/\sqrt{d_i} + y_i \sqrt{d_i}\right)^2. \tag{6}$$

Thus, $CG(i)$ can be calculated by the $n$-th column $(\mathbf{h}_i, d_i)$ of $(\mathbf{H}^{\infty})^{-1}$, without the need of calculating $(\mathbf{H}_{-i}^{\infty})^{-1}$ when $i = n$. The case for general $i \neq n$ can also be readily solved by permuting the $i$-th row and column of $\mathbf{H}^{\infty}$ into the last positions.

**Correlation to other scores** We show relation between our score and other data valuation scores that require the training of neural networks. Toneva et al. (2019) define 'the forgetting score' for each training example as the number of times during training the decision of that sample switches from a correct one to incorrect one. Paul et al. (2021), on the other hand, suggest the GraNd score, which is the expected loss gradient norm $\mathbb{E}[\|\nabla_{\mathbf{W}(k)} l(u(k), y)\|]$, to bound the contribution of each training example to the decrease of loss on any other example over a single gradient step. The GraNd score is further approximated (under some assumptions) by the EL2N score, defined to be $\mathbb{E}[\|y - u(k)\|]$ where $u(k)$ is the output of the neural network for the sample $(x, y)$ at the $k$-th step. Since $|y - u(k)|$, if rescaled, is an upper bound on 0–1 loss, $\sum_k |y - u(k)|$ upper bounds forgetting score after rescaling. Thus, an example with a high forgetting score will also have a high GraND score and high EL2N score averaged over multiple time steps. We next relate our complexity-gap score to $\sum_k (y - u(k))$, and thus to all the three previous scores defined using training dynamics.

It was shown in (Arora et al., 2019) that for the overparameterized two-layer networks trained by GD, the gap between the label vector and the network output at the step $k$ can be approximated as $\mathbf{y} - \mathbf{u}(k) \approx (I - \eta\mathbf{H}^{\infty})^k \mathbf{y}$. Thus, we get $\sum_{k=0}^{\infty} \mathbf{y} - \mathbf{u}(k) \approx \sum_{k=0}^{\infty} (I - \eta\mathbf{H}^{\infty})^k \mathbf{y} = \frac{1}{\eta}(\mathbf{H}^{\infty})^{-1}\mathbf{y}$.

Table 1: Spearman's rank correlation between CG-score (CG'-score) and other data valuation scores.

| Datasets | Correlation btwn. CG-score and | | | Correlation btwn. CG'-score and | | |
|---|---|---|---|---|---|---|
| | C-score | Forgetting | EL2N | C-score | Forgetting | EL2N |
| CIFAR-10 | 0.557 | 0.432 | 0.365 | 0.115 | 0.110 | 0.136 |
| CIFAR-100 | 0.529 | 0.289 | 0.356 | 0.243 | 0.090 | 0.177 |

Without loss of generality, consider $i = n$. Then, the accumulated difference between $y_i$ and $u_i(k)$ over $k$ can be approximated as

$$\sum_{k=0}^{\infty} (y_i - u_i(k)) \approx \left( \mathbf{h}_i^\top \mathbf{y}_{-i} + y_i d_i \right) / \eta = \left( \mathbf{y}_{-i}^\top \mathbf{h}_i / \sqrt{d_i} + y_i \sqrt{d_i} \right) \sqrt{d_i} / \eta. \tag{7}$$

Note that both the right-hand side of equation 7 and $\mathrm{CG}(i)$ in equation 6 depend on the term $\left( \mathbf{y}_{-i}^\top \mathbf{h}_i / \sqrt{d_i} + y_i \sqrt{d_i} \right)$. Thus, we can expect that $\mathrm{CG}(i)$ will be correlated with the scores related to training dynamics, including the forgetting score, GraND score and EL2N score. Different from those scores, our score can be directly calculated from the data without training of a model.

**Inversion of $\mathbf{H}^\infty$ is an effective step**  In the definition of $\mathrm{CG}(i)$ in equation 2, we use the inverse of $\mathbf{H}^\infty$ to measure the alignment of the eigenvectors of $\mathbf{H}^\infty$ with the label vector $\mathbf{y}$. One could suggest another score that directly uses $\mathbf{H}^\infty$ instead of $(\mathbf{H}^\infty)^{-1}$, which can save the computation for inversion. However, we argue that the score calculated by using $(\mathbf{H}^\infty)^{-1}$ includes more information than the score calculated by $\mathbf{H}^\infty$ due to the reason described below. Let us define $\mathrm{CG}'(i) := \mathbf{y}^\top \mathbf{H}^\infty \mathbf{y} - \mathbf{y}_{-i}^\top \mathbf{H}_{-i}^\infty \mathbf{y}_{-i}$. Without loss of generality, assume $i = n$. Then, $\mathrm{CG}'(i) = 2y_i(\mathbf{g}_i^\top \mathbf{y}_{-i}) + y_i^2 c_i$ where $\mathbf{g}_i = (\mathbf{H}^\infty)_{1:(n-1),n}$ and $c_i = H_{n,n}^\infty$ as defined in equation 3. Since $c_i = 1/2$ for all $i \in [n]$ from the definition of $\mathbf{H}^\infty$ in equation 1 and $y_i^2 = 1$ for $y_i = \pm 1$, we have $\mathrm{CG}'(i) = 2(y_i(\mathbf{H}^\infty \mathbf{y})_i - 1/2) + 1/2$. By using the approximation $\mathbf{y} - \mathbf{u}(k) \approx (I - \eta \mathbf{H}^\infty)^k \mathbf{y}$ from Arora et al. (2019), we have $\mathbf{y} - \mathbf{u}(1) \approx \mathbf{y} - \eta \mathbf{H}^\infty \mathbf{y}$ when $k = 1$, which implies $\mathbf{u}(1) \approx \eta \mathbf{H}^\infty \mathbf{y}$. Thus, $\mathrm{CG}'(i) = \frac{2}{\eta} y_i u_i(1) + 1/2$. Note that an instance $(\mathbf{x}_i, y_i)$ has a large $\mathrm{CG}'(i)$ if $y_i u_i(1)$ is large, i.e., the network output $u_i(1)$ after 1-time step has the same sign as the targeted label $y_i \in \{-1, 1\}$ and has a large magnitude. Thus, $\mathrm{CG}'(i)$ measures how fast the training instance can be learned by the neural network. Different from $\mathrm{CG}'(i)$, our original score $\mathrm{CG}(i)$ is correlated with the accumulated error between $y_i$ and $u_i(k)$ averaged over the training as shown in equation 7. Thus, our original score $\mathrm{CG}(i)$, using the inverse of $\mathbf{H}^\infty$, reflects the statistics averaged over the whole training steps.

In Table 1, we compare the Spearman'a rank correlation between CG-score/CG'-score and other data valuation scores including C-score (Jiang et al., 2021), forgetting score (Toneva et al., 2019) and EL2N score (Paul et al., 2021) for CIFAR-10/100 datasets.[1] We can observe that CG-score has higher correlations with the previous scores compared to those of CG'-score. In the rest of this paper, we focus on the CG-score for our data valuation score.

### 3.3 GEOMETRIC INTERPRETATION OF THE COMPLEXITY-GAP SCORE

We next provide geometric interpretation for the CG-score. For the sake of simplicity, we consider binary dataset with $\frac{n}{2}$ samples having $y_i = 1$ and $\frac{n}{2}$ samples having $y_i = -1$. We further assume that $\mathbb{E}[H_{ij}^\infty] = p$ if $y_i = y_j$ and $\mathbb{E}[H_{ij}^\infty] = q$ if $y_i \neq y_j$ for some $|p| > |q|$. Note that the diagonal entires $H_{ii}^\infty = 1/2$ for all $i \in [n]$. Thus, $\mathbb{E}[\mathbf{H}^\infty]$ can be decomposed as $\mathbb{E}[\mathbf{H}^\infty] = \left( \frac{1}{2} - p \right) \mathbf{I}_n + \mathbf{S}$ where $\mathbf{S}$ is a block matrix with $\mathbf{S} = \begin{pmatrix} p & q \\ q & p \end{pmatrix} \otimes \mathbf{I}_{n/2}$, and the resulting eigenvalues of $\mathbb{E}[\mathbf{H}^\infty]$ are $\frac{(p+q)n}{2} + \left( \frac{1}{2} - p \right)$, $\frac{(p-q)n}{2} + \left( \frac{1}{2} - p \right)$ and $\left( \frac{1}{2} - p \right)$ with multiplicity $(n-2)$. When $p = \Theta(q)$ and $p = o(1/n)$, the matrix $\mathbb{E}[\mathbf{H}^\infty] \approx (1/2)\mathbf{I}_n$. From the representations of $\mathbf{H}^\infty$ and $(\mathbf{H}^\infty)^{-1}$ in equation 3 and the implied relation $\mathbf{h}_i^\top = -c_i^{-1} \mathbf{g}_i^\top (\mathbf{H}^\infty)_{n-1}^{-1}$, assuming $\mathbf{H}^\infty \approx (1/2)\mathbf{I}_n$ and using $c_i = 1/2$, we can write $\mathbf{h}_i^\top = -4\mathbf{g}_i^\top$. By using this approximation, our CG-score in equation 6 can

---

[1]The way we calculate CG-scores for multi-label datasets is explained in Appendix §A. To reduce the computation complexity, we calculated the scores by sub-sampling data and averaging them over multiple runs.

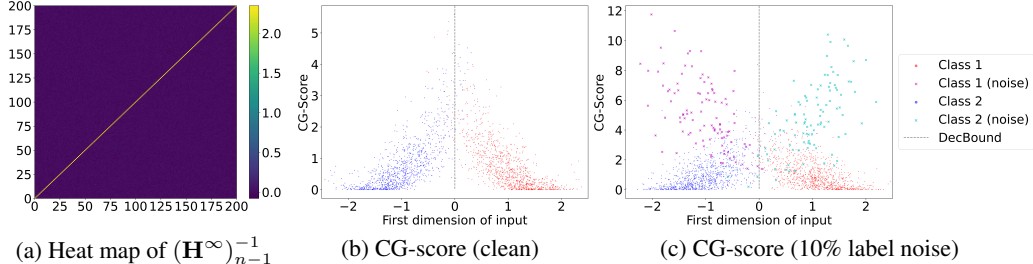

Figure 1: (a) Heat map of $(\mathbf{H}^\infty)_{n-1}^{-1}$ for 200 indices. We plot 200 indices only for clear visualization. (b) Scatter graph of CG-Score for two groups of samples from 3000-D Gaussian distributions having the same mean except the first dimension, where class 1 has mean +1 (red) and class 2 has mean -1 (blue). Samples near the boundary ($x_1 = 0$) tend to have higher CG-score. (c) Same plot as (b) with 10% label noise. Samples with label noise (marked by plus symbol) tend to have higher CG-score.

be approximated as

$$\mathrm{CG}(i) = d_i^{-1}(\mathbf{y}_{-i}^\top \mathbf{h}_i)^2 + 2y_i(\mathbf{y}_{-i}^\top \mathbf{h}_i) + y_i^2 d_i \approx 8(y_i(\mathbf{y}_{-i}^\top \mathbf{g}_i))^2 - 8y_i(\mathbf{y}_{-i}^\top \mathbf{g}_i) + 2 \qquad (8)$$

since $d_i \approx 2$ and $y_i^2 = 1$. Since $y_i(\mathbf{y}_{-i}^\top \mathbf{g}_i) = \sum_{j \in \{[n]: y_i = y_j\}} H_{ij}^\infty - \sum_{j \in \{[n]: y_i \neq y_j\}} H_{ij}^\infty$ and thus $\mathbb{E}[y_i(\mathbf{y}_{-i}^\top \mathbf{g}_i)] = \frac{(p-q)n}{2} = o(1)$, assuming $p = \Theta(q)$ and $p = o(1/n)$, the second term $-8y_i(\mathbf{y}_{-i}^\top \mathbf{g}_i)$ becomes much larger than the first term, which is proportional to $(y_i(\mathbf{y}_{-i}^\top \mathbf{g}_i))^2$. Thus, $-8y_i(\mathbf{y}_{-i}^\top \mathbf{g}_i)$ is the main term that determines the order of $\{\mathrm{CG}(i)\}_{i=1}^n$. Note that $y_i(\mathbf{y}_{-i}^\top \mathbf{g}_i)$ measures the gap between $\sum_{j \in \{[n]: y_i = y_j\}} H_{ij}^\infty$ and $\sum_{j \in \{[n]: y_i \neq y_j\}} H_{ij}^\infty$. Since $H_{ij}^\infty = \frac{\mathbf{x}_i^\top \mathbf{x}_j (\pi - \arccos(\mathbf{x}_i^\top \mathbf{x}_j))}{2\pi}$, $y_i(\mathbf{y}_{-i}^\top \mathbf{g}_i)$ measures the gap between the average similarities of $\mathbf{x}_i$ with other samples $\{\mathbf{x}_j\}$ of the same class $y_j = y_i$ and that with samples of the different class $y_j \neq y_i$, where the similarity is measured by the cosine between the two samples $(\mathbf{x}_i, \mathbf{x}_j)$ multiplied by the chance that ReLU is activated for both the samples. Thus, we can expect two important trends regarding the CG-score:

1. Regular vs. irregular samples: If a sample $(\mathbf{x}_i, y_i)$ is a 'regular' sample representing the class $y_i$ in the sense that it has a larger similarity with other samples of the same class than to samples from the different class, then it will have a large $y_i(\mathbf{y}_{-i}^\top \mathbf{g}_i)$, resulting in a lower CG-score due to the minus sign. On the other hand, if a sample $(\mathbf{x}_i, y_i)$ is 'irregular' in the sense that it does not represent the class and has a small similarity with the samples of the same class, then $y_i(\mathbf{y}_{-i}^\top \mathbf{g}_i)$ will be small, resulting in a higher CG-score.

2. Clean label vs. noisy label: A sample with label noise tends to have a large CG-score since the order of two terms in subtraction at $y_i(\mathbf{y}_{-i}^\top \mathbf{g}_i) = \sum_{j \in \{[n]: y_i = y_j\}} H_{ij}^\infty - \sum_{j \in \{[n]: y_i \neq y_j\}} H_{ij}^\infty$ is effectively switched for label-noise samples. Thus, $y_i(\mathbf{y}_{-i}^\top \mathbf{g}_i)$ tend to be negative for a sample with label noise, while it is positive for clean-label samples.

We empirically demonstrate the above two trends by a controlled experiment. Consider two 3000-dimensional Gaussian distributions with a fixed covariance matrix $0.25I_{3000}$ and different means $(1, 0, \ldots 0)$ and $(-1, 0, \ldots, 0)$, representing class +1 and -1, respectively. We generate 1000 samples from each class. We first check whether the structural assumptions on $\mathbf{H}^\infty$ hold with this synthetic dataset. In Fig. 1a, we observe that $(\mathbf{H}^\infty)_{n-1}^{-1}$ can be well approximated by $2I_{n-1}$, and thus the approximation for $\mathrm{CG}(i)$ in equation 8 may hold well. We then examine the correlation between the CG-score and the sample value at the first dimension. In Fig. 1b, we can observe that instances near the decision boundary $x_1 = 0$ have higher CG-scores. This shows that irregular samples, e.g., samples near the boundary of the two classes, indeed have higher CG-scores. We further modify the generated samples by randomly flipping the labels of 10% samples and re-calculate the CG-scores of all the samples. As shown in Fig. 1c, the samples with label noise tend to have higher CG-scores, agreeing with our intuition. In Appendix §B, we further discuss discriminating mislabeled data and atypical (but useful) data by using CG-score, and also show high/low-scoring examples for public datasets including MNIST, FMNIST, CIFAR-10 and ImageNet in Appendix §C.

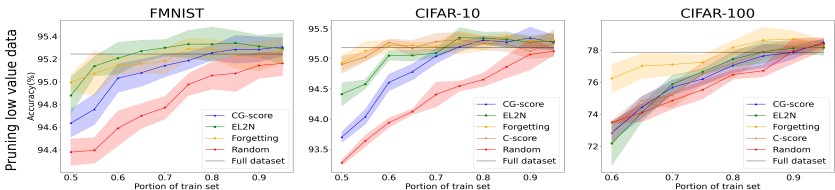

(a) Pruning low-scoring examples first. Better score maintains the test accuracy longer.

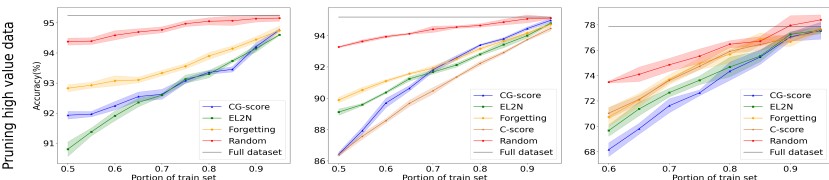

(b) Pruning high-scoring examples first. Better score makes the rapid performance drop.

Figure 2: Pruning experiments with FMNIST (left), CIFAR-10 (middle) and CIFAR-100 (right). Only CG-Score does not require any training of the model to calculate the scores, but it achieves competitive performances and outperforms the random baseline. In (a), CG-score maintains the test accuracy up to significant removing portions; in (b), the test accuracy drops most rapidly for CG-score since the examples with high CG-score are essential in generalization of the model.

## 4 DATA VALUATION THROUGH COMPLEXITY GAP SCORE

### 4.1 DATA PRUNING EXPERIMENTS

To evaluate the ability of the CG-score in identifying important examples, we design data pruning experiments, similar to those in Ghorbani & Zou (2019); Paul et al. (2021). We evaluate our score on three public datasets, FMNIST, CIFAR-10/100 and train ResNet networks (He et al., 2016), ResNet18 for FMNIST and CIFAR-10 and ResNet50 for CIFAR-100 dataset, respectively. As baseline methods for data valuation, we use three state-of-the-art scores, C-score (Jiang et al., 2021), EL2N (Paul et al., 2021), and Forgetting score (Toneva et al., 2019), all of which require training of models for computation. On the contrary, our score is a data-centric measure and independent on the model. More details on the baselines and experiments are summarized in Appendix §D.

In Figure 2, the first row shows the test accuracy of the model trained with different subset sizes when low-scoring (regular) examples are removed first, while the second row shows the similar result but when high-scoring (irregular) examples are removed first. We report the mean of the result after five independent runs, and the shaded regions indicate the standard deviation. The gray line (full dataset) represents the results when all data is used, while the red curve (random) represents the results when examples are randomly removed. We can observe that when removing low-scoring examples first, networks maintain the test accuracy up to significant removing portions, i.e., 40% for CIFAR-10 and 20% for CIFAR-100 with less than 1% test accuracy drop. Our CG-score, which does not require any training of a model, can achieve competitive performances as the other baseline methods and it also significantly outperforms the random baseline. When removing the high-scoring examples first, the test accuracy drops most rapidly for the CG-score curve, implying that examples with the high CG-score are essential part of the data governing the generalization of the model.

We show the effectiveness of CG-score in more complicated models and also in fine-tuning by reporting the data pruning experiments at DenseNetBC-100 (Huang et al., 2017) and Vision Transformer (ViT) (Dosovitskiy et al., 2021) in Appendix §E. We also compare CG-score with two more baselines, TracIn (Pruthi et al., 2020) and CRAIG (Mirzasoleiman et al., 2020), in Appendix §F.

### 4.2 DETECTING LABEL NOISE

In Section 3.3, we analyzed the CG-score and showed that the examples with label noise tend to have higher CG-score. We next empirically investigate the ability of the CG-score in detecting label noise

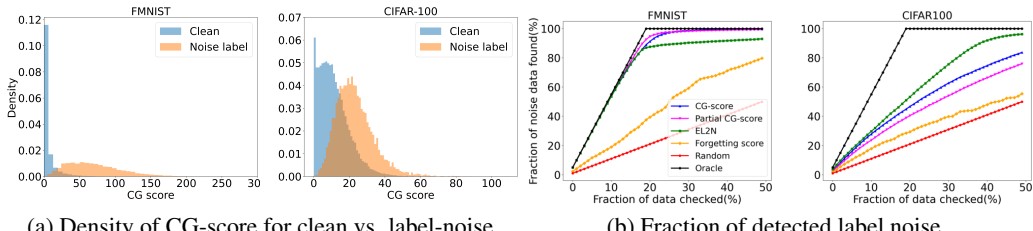

(a) Density of CG-score for clean vs. label-noise      (b) Fraction of detected label noise

Figure 3: (a) Density of CG-score for clean (80%) vs. label-noise (20%) examples. Examples with label noise tend to have higher CG-score. For FMNIST (left) dataset, the CG-score histograms betwen clean and label-noise groups are better separated than those for CIFAR-100 (right). (b) Fraction of label noise ($y$-axis) included in the examined portion ($x$-axis) for 20% label noise. CG-score (blue) achieves better noise detectability for FMNIST than for CIFAR-100.

by artificially corrupting 20% of instances in FMNIST and CIFAR-100 with random label noise. We first examine the distribution of the CG-scores for each dataset after the corruption in Figure 3a. We can observe that for both datasets, the examples with corrupted labels (orange) tend to have higher CG-scores than those with clean labels (blue). For a relatively simpler dataset, FMNIST (left), the CG-score histograms can be more clearly separable between the clean and noisy groups, compared to a more complex dataset, CIFAR-100 (right). With this observation, we can anticipate that the CG-score can have a better detectability of label noise for relatively simpler datasets.

We evaluate the noise detectability of the CG-score by checking a varying portion of the examples sorted by the CG-score (highest first) in Fig. 3b. The curves indicate the fraction of noisy examples included in the examined subset. We compare our score with two other scores, Forgetting score and EL2N, as well as a random baseline and the oracle. We can observe that the CG-score achieves the best performance, near that of the oracle, for FMINST, and competitive performances for CIFAR-100. In the plot, we also compare the performance of the CG-score with that of 'Partial CG-score,' which is a new score defined by a sub-term of the CG-score. In the CG-score in equation 6, we have three terms, but only the second term $2y_i(\mathbf{y}_{-i}^\top \mathbf{h}_i)$ uses the label information of the data $(\mathbf{x}_i, y_i)$. Thus, we examined the ability of this term only, defined as the 'Partial CG-score', in detecting the label noise, and found that the CG-score and partial CG-score have similar performances in detecting label noise, which implies that the label-noise detectability of the CG-score mainly comes from the term $2y_i(\mathbf{y}_{-i}^\top \mathbf{h}_i)$. More experiments with higher noise rate are reported in Appendix §G.

## 5 COMPLEXITY-GAP SCORE AND DEEP LEARNING PHENOMENA

We next demonstrate that the CG-score can capture 'learning difficulty' of data instances.

### 5.1 COMPLEXITY GAP SCORE CAPTURES LEARNING DIFFICULTY

It has been widely observed that there exists an order of instances in which a model learns to classify the data correctly and this order is robust within model types (Wu et al., 2021). Many previous scoring functions use this type of observation and define the 'difficulty' of a given instance based on the speed at which the model's prediction converges (Toneva et al., 2019; Swayamdipta et al., 2020). Our data-centric CG-score is originally defined based on the gap in the generalization error bounds as in equation 2, but it also reflects the convergence speed of the instance, as analyzed in equation 7. We empirically demonstrate this relation by analyzing the training dynamics of CIFAR-10 dataset trained in ResNet18 network. We first sort the data instances in ascending order using the CG-score, and divide the data into 10 equal-sized subgroups. We then measure the mean of loss and training accuracy for the 10 subgroups as training progresses. Fig. 4a and Fig. 4b show the mean of loss and the training accuracy throughout the training for the 10 subgroups, respectively, where the blue color represents the low-scoring groups and the red color represents the high-scoring groups. We can observe that the mean loss and accuracy converge faster for low-scoring groups, while it takes longer to converge for high-scoring groups. This indicates that the CG-score is highly correlated with the 'difficulty' of an example measured by the learning speed at a model.

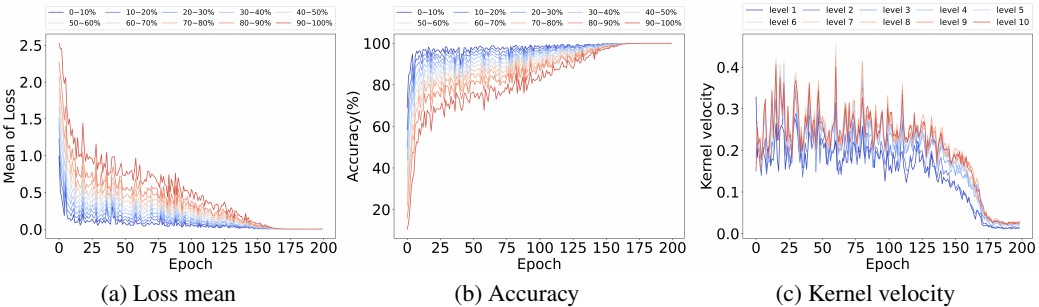

(a) Loss mean     (b) Accuracy     (c) Kernel velocity

Figure 4: Training dynamics measured for each subset of CIFAR-10 examples, grouped by the CG-score. The line color varies over groups, where the blue lines include low-scoring examples and the red lines include high-scoring groups. (a) and (b) Mean loss and accuracy for subgroups of CIFAR-10 data, sorted by the CG-score, trained on ResNet18. The mean loss and accuracy converge slowly for high-scoring groups (red lines). (c) Kernel velocity of CIFAR-10 examples grouped by the CG-score. The Kernel evolves with a higher velocity for high-scoring groups throughout the training.

## 5.2 DATA SAMPLES DRIVING MOVEMENT OF NEURONS

We next investigate the relation between the CG-score and the evolution velocity of the data-dependent Neural Tangent Kernel (NTK) (Fort et al., 2020). NTK has been widely used to approximate the evolution of an infinite-width deep learning model via linearization around initial weights, when the network is trained by gradient descent with a sufficiently small learning rate (Jacot et al., 2018). The main idea is that in the limit of an infinite width, the network parameters do not move very far from its initialization throughout the training, so that the learning process can be approximated by a linear process along the tangent space to the manifold of the model's function class at the initialization (Lee et al., 2019a). However, as observed in Fort et al. (2020), for a finite-width network, the tangent kernel is not constant but it rather rapidly changes over time, especially at the beginning of the training. To quantify this change, Fort et al. (2020) addressed the data-dependent Kernel Gram matrix, which is the Gram matrix of the logit Jacobian, and defined the Kernel velocity as the cosine distance between two NTK Gram matrices before and after one epoch of training. In Paul et al. (2021), the Kernel velocity was used to evaluate the subgroups of data instances to figure out the subgroup driving the learning and the change in the NTK feature space. We conduct similar experiments for subgroups of data, divided according to our CG-score. Fig. 4c shows the Kernel velocities for 10 different subgroups of CIFAR-10 data trained in ResNet18. Each group is composed of 125 consecutive instances from each level, where the level is defined by dividing the full dataset, sorted in ascending order by the CG-score, into 10 groups. The higher level (red) is composed of instances having higher CG-scores, while the lower level (blue) is composed of instances having lower CG-scores. We can observe that the samples with high CG-score (red) maintains higher Kernel velocity throughout the training, which means that NTK Gram matrix evolves by a larger amount for the samples of high CG-score. Thus, we can hypothesize that the instances with high CG-score are 'difficult' examples the network may struggle to optimize and try to fit throughout the training.

## 6 DISCUSSION

We proposed the CG-score, a data-centric valuation score, to quantify the effect of individual data instances in optimization and generalization of overparameterized two-layer neural networks trained by gradient descent. We theoretically and empirically demonstrated that the CG-score can identify 'irregular' instances within each class, and can be used as a score to select instances essential for generalization of a model or in filtering instances with label noise. We also showed the close relation between the CG-score and learning difficulty of instances by analyzing training dynamics. Interesting open problems related to the CG-score include 1) providing theoretical justification of the score for more general deep neural networks and 2) improving the score by modifying the definition in terms of 'features' of data instances. In Appendix §H, we provide further discussion for 'feature-space CG-score' and report experimental results, demonstrating the effectiveness of the score.

## ACKNOWLEDGEMENT

This research was supported by the National Research Foundation of Korea under grant 2021R1C1C11008539, and by the Ministry of Science and ICT, Korea, under the IITP (Institute for Information and Communications Technology Panning and Evaluation) grant No.2020-0-00626.

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

## A    EXTENSIONS TO MULTI-CLASS COMPLEXITY GAP SCORE

In this section, we explain the details of how we calculate the CG-scores for multi-label public datasets.

### A.1    CALCULATION OF CG-SCORE FOR MULTI-LABEL DATASETS

In defining the CG-score as equation 2, we assumed the binary datasets $y \in \{\pm 1\}$ with inputs having a fixed norm $\|\mathbf{x}\|_2 = 1$. To calculate the CG-score for multi-label ($k$-class) public datasets, we first normalize all the inputs to have $\|\mathbf{x}\|_2 = 1$. We then calculate the CG-score for examples of each class $j \in [k]$, assuming that all the examples from class $j$ have label $+1$ and the rest of the examples from any other classes have label $-1$. In detail, let $\mathbf{y} = (y_1, y_2, ..., y_n) \in \{1, 2, ... k\}^n$ be the label vector for $n$ data instances and, without loss of generality, assume that $\mathbf{x}_1, \mathbf{x}_2, \ldots, \mathbf{x}_l$ belong to class 1, i.e. $y_1, y_2, ..., y_l = 1$. Then, to calculate the CG-score for $\mathbf{x}_1, \mathbf{x}_2, \ldots \mathbf{x}_l$, we generate the gram matrix $\mathbf{H}^\infty$ with $(\mathbf{x}_1, 1), (\mathbf{x}_2, 1), \ldots, (\mathbf{x}_l, 1), (\mathbf{x}_{l+1}, -1), (\mathbf{x}_{l+2}, -1), \ldots, (\mathbf{x}_n, -1)$ and calculate the CG-scores of $\mathbf{x}_1, \mathbf{x}_2, \ldots, \mathbf{x}_l$ as the two-label case.

### A.2    STOCHASTIC METHOD TO CALCULATE CG-SCORES

Since the calculation of the CG-score requires taking the inverse of a $n \times n$-dimensional matrix $\mathbf{H}^\infty$ where $n$ is the number of total samples, it would demand expensive memory and computational cost for large $n$. To lower the complexity, we sub-sample the examples with class $-1$ so that the ratio between class +1 and class -1 is reduced from $1 : (k-1)$ to $1 : 3$ for MNIST and FMINST, $1 : 4$ for CIFAR-10 and CIFAR-100. We repeat this process 10 times by randomly sampling examples of label $-1$ and then average out the calculated CG-score of each example from class $+1$.

### A.3    JUSTIFICATION OF STOCHASTIC METHOD TO CALCULATE CG-SCORES

To justify the stochastic method in calculating the CG-scores, we conduct an experiment to check whether the CG-scores calculated by the stochastic method converges well to the true CG-scores utilizing the full dataset. We created a subset of the CIFAR-10 dataset, the Small-CIFAR-10 dataset, which consists of 1,000 instances for each label (a total of 10,000 instances). Then, we compared the CG-score calculated by 10,000x10,000 Gram matrix $\mathbf{H}^\infty$ of the full dataset (true CG-score) with the CG-score calculated by the stochastic method, where the stochastic CG-score for the instances from a class (class 1) are calculated by subsampling the samples from any other classes (class -1) with the ratio between class 1 and -1 equal to $1 : 1$ (size 2,000x2,000), $1 : 2$ (size 3,000x3,000), $1 : 3$ (size 4,000x4,000) and $1 : 4$ (size 5,000x5,000) instead of $1 : 9$. We repeat this process multiple times by randomly sampling examples of label $-1$ and then average out the calculated CG-score of each example from class $+1$.

In Figure 5, we plot the Spearman's rank correlation and Pearson correlation between the true CG-score and the stochastic CG-score for each ratio (different colors) as the number of random sampling increases. We can observe that the correlations converge to a certain number as the number of random sampling increases, and the amount of correlation increases as the size of the matrix (the number of instances included in defining $\mathbf{H}^\infty$) increases. The values of correlations obtained after 20 independent runs are 0.829, 0.914, 0.953, and 0.973 for Spearman rank correlations and 0.796, 0.898, 0.943, and 0.967 for Pearson rank correlations.

**Recommended number of runs for stochastic calculation.**    The proper number of runs in stochastic calculation of the CG-score may need to be determined by the size and the number of classes of the datasets. We recommend the number of runs large enough to include at least a half of the whole dataset in calculating the CG-score for samples of each class. As an example, for the CIFAR-100 dataset, where each class includes 500 images, when the sampling ratio between the class of interest and the rest of classes is 1:4, we calculate the score for 500 images from the class of interest by using 2,000 images from the rest of 99 classes. Then, about 20 images are selected from each of the 99 classes. To cover at least a half of the images per class (250), we need to repeat the runs about 10 times (ignoring overlap of samples in each run). For the FMNIST/CIFAR-10 datasets, the similar calculation shows that only 2-3 runs will be enough.

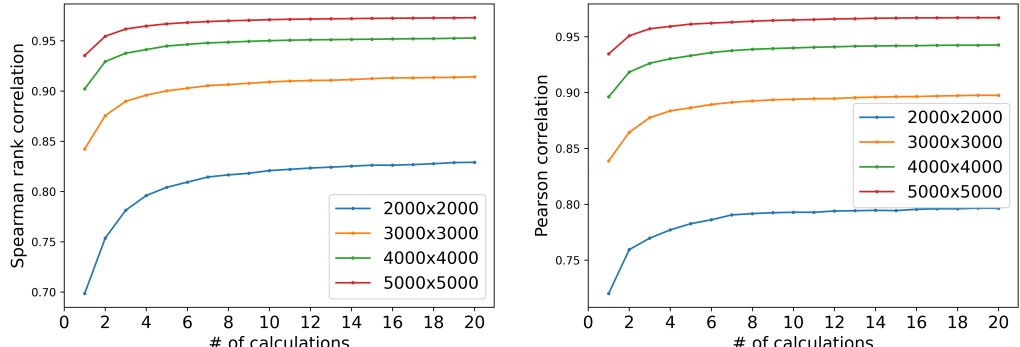

Figure 5: We create Small CIFAR-10 dataset by sampling 1,000 data from each class of CIFAR-10 dataset. Figures show Spearman's rank correlation(left) and Pearson correlation(right) between the CG-score of Small CIFAR-10 and CG-score calculated by subsampling the dataset (stochastic CG-score) as the number of calculations (random sampling) increases. Stochastic CG-scores are calculated with 2,000 (cyan), 3,000 (orange), 4,000 (green), and 5,000 (red) instances, while the full data includes 10,000 instances (1,000 for each class). X-axis indicates the number of independent runs to calculate averaged score, and Y-axis indicates the correlations.

# B    DISCRIMINATING MISLABELED DATA FROM ATYPICAL (BUT USEFUL) DATA BY CG-SCORES

In Section 3.3, we showed that irregular samples, either from input itself or mislabelling, tend to have high CG-scores. Discriminating mislabeled data and atypical (but useful) data is a major challenge in data valuation, since both the mislabeled data and atypical data are irregular in the data distribution and tend to have high CG-scores. The same challenge has been observed with previous valuation scores such as forgetting score (Toneva et al., 2019) and EL2N score (Paul et al., 2021).

However, we find that mislabeled data usually has a higher CG-score than atypical data, and this tendency gives us the possibility to separate mislabeled data from the rest of the clean data. To check the tendency, we perform a data window experiment. The data instances are sorted in ascending order by the CG-score, and we compare the test accuracy of a neural network trained with 50% of training instances selected from an offset% to (offset+50)% scoring group, for different offset points of $\{0, 5, 10, \ldots, 45, 50\}$. For example, when the offset is 20%, we select the data instances from 20% to 70% scoring examples. When the training instances do not include mislabeled data, we can expect that the window experiment will show higher accuracy as the offset increases up to 50%. We add 20% of random label noise to FMNIST and CIFAR-10 datasets to see how the trend changes when the dataset includes mislabeled instances.

As shown in Fig. 6 (a) and (b), the test accuracy increases until the offset reaches 30% and then drops after the point. When the offset is 30%, the 50%-width window includes 30% to 80% scoring examples. Since 20% mislabeled data are mainly located within the 80% to 100% scoring group, as the offset increases above 30%, the 50%-width window starts to include mislabeled instances and this causes the rapid drop of the test accuracy. Thus, from the window experiments we can see that the mislabeled data has the highest CG-score and can be separable from the rest of the clean examples by the CG-score.

Then, the next reasonable question is how to set the threshold on CG-score to detect the mislabeled data when the portion of mislabeled data is unknown. We show that the sign of the partial CG-score $2y_i(\mathbf{y}_{-i}^{\top}\mathbf{h}_i)$, which is a sub-term of the CG-score in equation 6 including the label information $y_i$, can be used in this purpose. As explained in Sec. 3.3, the partial CG-score measures the gap between the average similarity of the data instance with other samples of the same class compared to that with the samples of the different classes. Thus, by checking the sign of the partial CG-score, we can discriminate mislabeled samples from clean samples. In Fig. 6 (c), we show the scatter plot of clean (blue) and mislabeled data (orange), where one can find that mislabeled examples tend to have positive partial CG-score (x-axis). As shown in Table 2, we can check that for FMNIST with

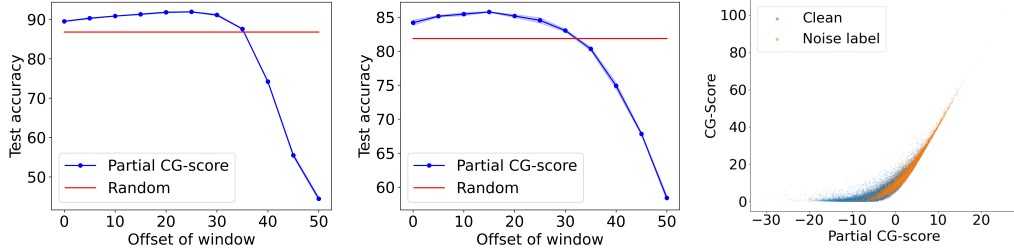

(a) Window experiment (FMNIST)  (b) Window experiment (CIFAR-10)  (c) Scatter graph (CIFAR-10)

Figure 6: (a) and (b) Window experiments with 20% label noisy for FMNIST (a) and CIFAR-10 (b) datasets. Test accuracy (y-axis) of a model trained with 50% of training instances selected from offset% (x-axis) to (offset+50)% scoring group. (c) Scatter graph of Partial CG-score (x-axis) and CG-score (y-axis) for CIFAR-10 with 20% label noise.

Table 2: Number of mislabeled data and clean data in each subset selected based on the partial CG-score ($2y_i(\mathbf{y}_{-i}^\top \mathbf{h}_i)$) for FMNIST and CIFAR-10 datasets, where 20% of samples in each dataset are corrupted with label noise.

| Dataset | FMNIST | | | CIFAR-10 | | |
|---|---|---|---|---|---|---|
| $2y_i(\mathbf{y}_{-i}^\top \mathbf{h}_i)$ | positive | negative | high 20% | positive | negative | high 20% |
| mislabeled | 11796 | 204 | 10776 | 7146 | 2854 | 5619 |
| clean | 3462 | 44538 | 1224 | 8331 | 31669 | 4381 |

20% label noise (12,000 mislabeled instances and 48,000 clean instances), 98%(=11796/12000) of mislabeled data ends up having positive partial CG-score, while only 7%(=3462/48000) of clean data has positive partial CG-score. Thus, even when the portion of label noise is unknown, our partial CG-score can effectively detect the mislabeled data by the sign information. This tendency was less clear for CIFAR-10 dataset due to the increased dataset complexity, but still the tendency existed.

## C  VISUALIZATION OF EXAMPLES SORTED BY CG-SCORE

**Analysis of MNIST and FMINST by CG-score**  In Fig. 7, we show examples of MNIST (Fig. 7a) and FMNIST (Fig. 7b) images sorted by CG-score. The top/bottom three rows show the top-/bottom-ranked examples, respectively. We can observe that the examples with the lowest CG-score are regular examples representing each class and they look similar to each other, while the examples with the highest scores are irregular and they look different among themselves, among which we can identify (possibly) mislabeled instances, marked with red rectangles. Similarly, in CIFAR-10 images (Fig. 7c), we observe that low-scoring examples (bottom two rows) are regular ones sharing similar features representing the class while high-scoring examples (top two rows) are irregular ones.

**Analysis of CIFAR-100 by CG-score**  Fig. 8 shows the examples of CIFAR-100 dataset. Fig. 8a shows the means and standard variations of 100 classes in CIFAR-100 dataset. Among 100 classes, 'orange' and 'plain' classes have small CG-score means and variances, while 'bottle' and 'flatfish' classes have large CG-score means and variances. We display ten top-/bottom-ranked examples of these four classes, with the histograms of the scores in Fig. 8b.

**Analysis of ImageNet by CG-score**  To check the effectiveness of CG-score in analyzing more complicated dataset, we compute CG-score on the ImageNet dataset as well. The CG-score is computed with the sampling ratio of 1:4 by averaging the results from 10 independents runs. Fig. 9 shows the distribution of CG-score and shows examples of ImageNet dataset with CG-score his-

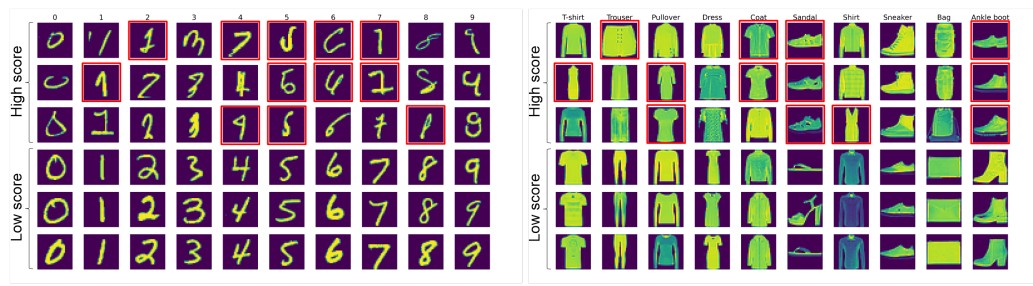

(a) Examples of MNIST dataset         (b) Examples of FMNIST dataset

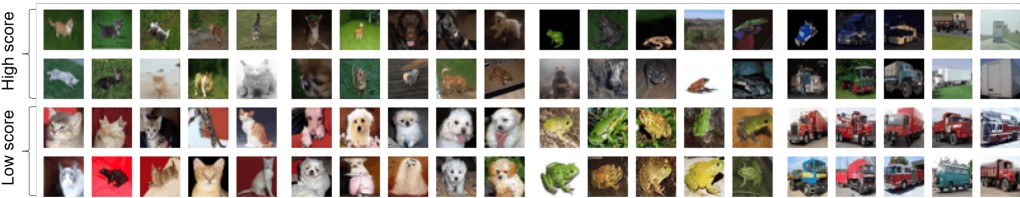

(c) Examples of CIFAR-10 dataset (Cat, dog, frog, and truck)

Figure 7: Examples sorted by CG-score. In (a) and (b), top-3 rows / bottom-3 rows display top-ranked / bottom-ranked examples, respectively. Among top-ranked examples, (possibly) mislabeled examples are marked by red rectangles. In (c), top-2 rows / bottom-2 rows display top-ranked / bottom-ranked examples, respectively.

togram. 'Jack-o-lantern' and 'Rapeseed' classes have relatively smaller mean and std of CG-score and the examples from these classes share some typical attributes (color and shape). On the other hand, 'Ladybug' has high standard deviation and we can observe clear differences between the low-scoring examples and high-scoring examples. Examples of 'Spatula', which has the biggest mean and relatively high standard deviation, do not look similar to each other but rather diverse. From the analysis, we can see that our CG-score is effective in examining high-resolution complicated dataset such as ImageNet, and the score reflects the instance-wise structural regularities, which can be used in analyzing or improving learning algorithms.

## D    IMPLEMENTATION DETAILS AND COMPUTATIONAL COST

### D.1    TRAINING DETAILS

In Section 4 and 5, we evaluate our score on three public datasets, FMNIST, CIFAR-10/100, by training ResNet networks (He et al., 2016) of different depths. ResNet18 is used for FMNIST and CIFAR-10 dataset and ResNet50 is used for CIFAR-100 dataset. Implementation of the ResNet is based on the ResNet network in torchvision (Paszke et al., 2019). Since FMNIST and CIFAR images are smaller than ImageNet (Deng et al., 2009) images, we replace the front parts of the ResNet (convolution layer with 7x7 kernel and 2x2 stride, max pooling layer with 3x3 kernel and 2x2 stride) with a single convolution layer with 3x3 kernel and 1x1 stride for small size image. The details on hyperparameters and optimization methods used in training are summarized in the Table 3.

### D.2    COMPUTATIONAL TIME

Table 4 provides computational time (in seconds) to obtain CG-scores with different sampling ratio 1:1, 1:2, 1:3, and 1:4 for FMNIST and CIFAR-10/100 dataset. We also report the time to compute the baseline scores, Forgetting (Toneva et al., 2019), EL2N (Paul et al., 2021), TracIn (Pruthi et al., 2020), and CRAIG (Pruthi et al., 2020). We do not calculate the CG-score of FMNIST dataset with sampling ratio 1:4 since it requires a inversion for 30,000x30,000 matrix, which exceeds the limit of

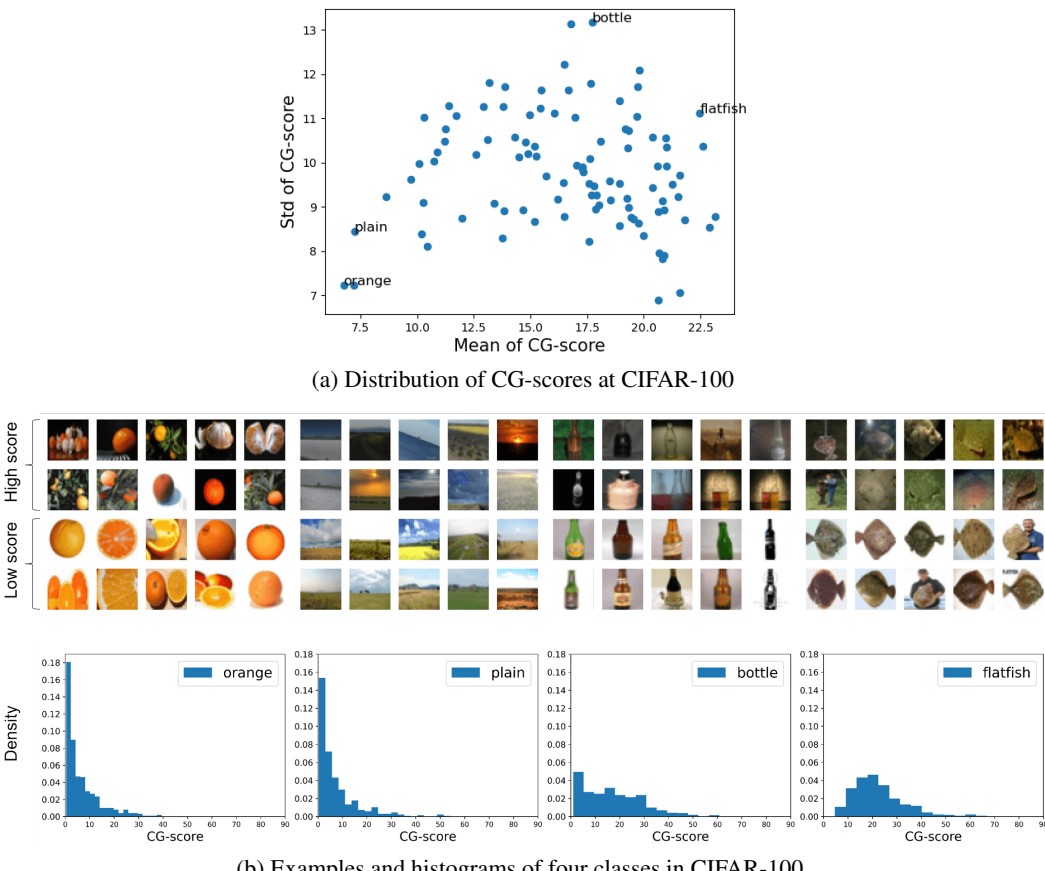

(a) Distribution of CG-scores at CIFAR-100

(b) Examples and histograms of four classes in CIFAR-100

Figure 8: Sample analysis for CIFAR-100 dataset in terms of the CG-score. (a) shows the CG-score means and standard deviations of 100 classes in CIFAR-100 dataset. (b) shows ten top-ranked examples (top two rows) and ten bottom ranked examples (bottom two rows) for orange, plain, bottle, and flatfish classes of CIFAR-100 dataset. Histograms show distributions of the CG-scores for each class.

Table 3: Details for the experiments used in the training of the dataset.

|  | FMNIST | CIFAR10 | CIFAR100 |
| --- | --- | --- | --- |
| Architecture | ResNet18 | ResNet18 | ResNet50 |
| Batch size | 128 | 128 | 128 |
| Epochs | 100 | 200 | 200 |
| Initial Learning Rate | 0.02 | 0.05 | 0.1 |
| Weight decay | 5e-4 | 5e-4 | 5e-4 |
| Optimizer | SGD with momentum 0.9 | | |
| Learning Rate Scheduler | Cosine annealing schedule (Loshchilov & Hutter, 2017) | | |
| Data Augmentation | Normalize by dataset's mean, variance Random Zero Padded Cropping (4 pixels on all sides) Random left-right flipping (probability 0.5) | | |

the device memory. Every score including ours and baselines needs to be calculated by averaging the results of independent multiple runs. For a fair comparison, we compare the time to get each score for a single run. Cost to compute CG-score depends on the size of the Gram matrix $\mathbf{H}^{\infty}$, since we need to calculate the inverse of $\mathbf{H}^{\infty}$ to get the CG-score and the computational complexity to

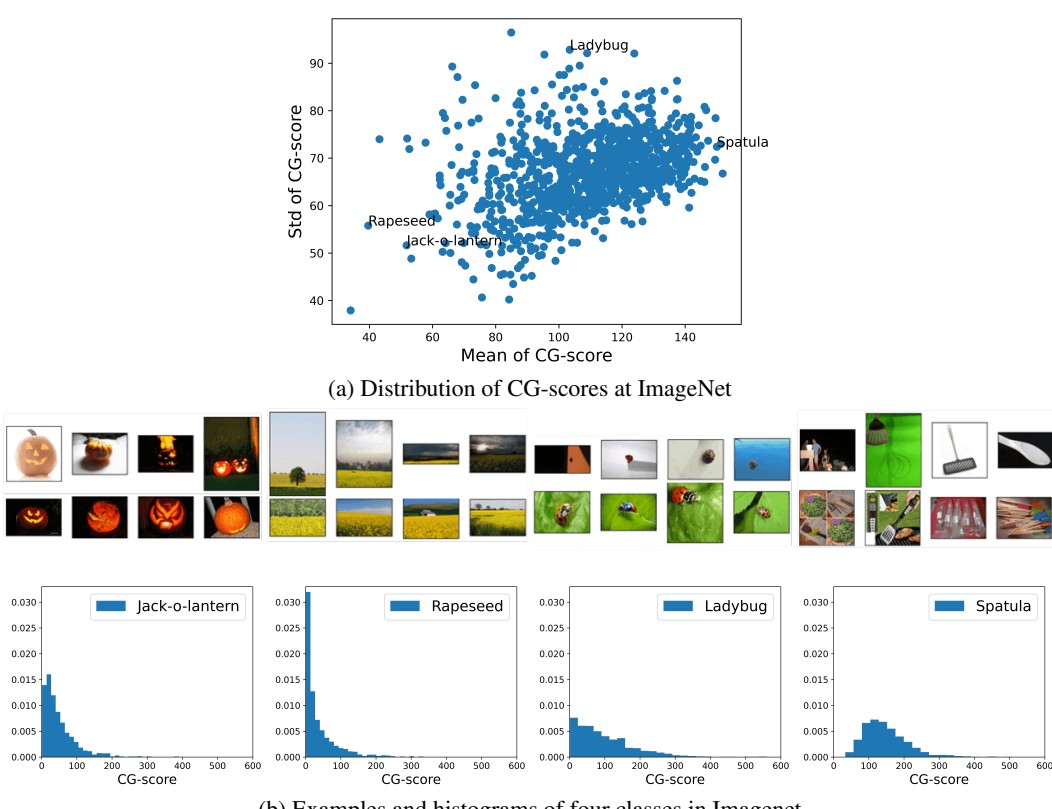

(a) Distribution of CG-scores at ImageNet

(b) Examples and histograms of four classes in Imagenet

Figure 9: Sample analysis for ImageNet in terms of the CG-score. (a) shows the CG-score means and standard deviations of 1,000 classes in ImageNet dataset. (b) shows four top-ranked examples (top row) and four bottom ranked examples (bottom row) for Jack-o-lantern, Rapeseed, Ladybug, and Spatula classes of ImageNet dataset. Histograms show density distributions of the CG-scores for each class.

Table 4: Time cost(seconds) to compute scores of the dataset.

|  | FMNIST | CIFAR-10 | CIFAR-100 |
| --- | --- | --- | --- |
| CG-score 1:1 | 1063 | 608 | 37.9 |
| CG-score 1:2 | 2610 | 1497 | 40.3 |
| CG-score 1:3 | 4834 | 2773 | 49.0 |
| CG-score 1:4 | - | 4388 | 61.1 |
| Forgetting | 1879 | 3322 | 6675 |
| EL2N | 370 | 323 | 662 |
| TracIn | 1856 | 3425 | 8400 |
| CRAIG | 3023 | 5263 | 10472 |
| GPU | | Nvidia A100 40GB | |

conduct an inversion of $n \times n$ matrix is $O(n^3)$. Therefore, for a dataset of which each class includes a large number of instances (e.g., FMNIST and CIFAR-10), taking the inverse of a Gram matrix with large sampling ratio may cause expensive computational cost, while computing the score for a dataset of which each class includes relatively small number of instances (CIFAR-100) can be done in a short time. For example, calculating CG-score of CIFAR-10 dataset (5,000 data in a class) takes 1.2 hours with sampling ratio 1:4, while that for CIFAR-100(500 data in a class) takes just 1 minute. EL2N score is time-efficient overall because it is calculate at the relatively early stage of training.

Forgetting and TracIn, on the other hand, take relatively longer time since they require at least one full train of the model. In addition, we argue that our method has another computational advantage that we do not need to search networks and hyperparameters which would work well for the target dataset.

## D.3 EXPERIMENTAL DETAILS

**Baseline scores for data valuation**  We use three state-of-the-art scores, C-score (Jiang et al., 2021), EL2N (Paul et al., 2021), and Forgetting score (Toneva et al., 2019) as baselines with which our CG-score is compared. We use pre-calculated C-score for CIFAR-10 and CIFAR-100 from Jiang et al. (2021) and calculate EL2N and Forgetting score by averaging the score across five independent training using the full dataset. We obtain EL2N scores at 20th epochs of the training. We use the same network architectures to calculate EL2N and Forgetting score: ResNet18 for FMNIST and CIFAR-10 dataset and ResNet50 for CIFAR-100 dataset. Detailed definitions of the scores are as follows:

- Consistency score (C-score): C-score of each instance is calculated by estimating the prediction accuracy of the instance attained by the model trained with the full dataset except the instance.

$$\text{C-score}(\mathbf{x}_i, y_i) = \mathbb{E}_n \left[ \hat{\mathbb{E}}^r_{S \sim \{(\mathbf{x}_j, y_j)\}^n_{j=1}} \left[ \mathbb{P}(f(\mathbf{x}_i; S \backslash \{(\mathbf{x}_i, y_i)\}) = y_i) \right] \right], \qquad (9)$$

  where $f(\mathbf{x}_i; S)$ is trained network using subset $S$, and $\hat{\mathbb{E}}^r$ denotes empirical averaging with $r$ i.i.d. samples of such subsets.

- Error L2-Norm (EL2N): The EL2N score of a training sample $(\mathbf{x}_i, y_i)$ is defined to be $\mathbb{E}[\|f(\mathbf{W}(t), \mathbf{x}_i) - y_i\|_2]$ where $f(\mathbf{W}(t), \mathbf{x})$ is the output of the neural network for the sample $(\mathbf{x}, y)$ at the $t$-th step.

- Forgetting score: Forgetting score is defined as the number of times during training (until time step $T$) the decision of that sample switches from a correct one to an incorrect one: $\text{Forgetting}(\mathbf{x}_i, y_i)$ is defined as

$$\sum_{t=2}^{T} \mathbb{1}\{\arg\max f(\mathbf{W}(t-1), \mathbf{x}_i) = y_i\}(1 - \mathbb{1}\{\arg\max f(\mathbf{W}(t), \mathbf{x}_i) = y_i\}). \qquad (10)$$

**Data pruning experiment**  We report the mean of the results after five independent runs. The shaded regions indicate the standard deviation. We acquire each result by training a network using a dataset pruned by specified portions, where the training instances are ordered by each score. We compute the number of iterations at which all data can be used in one epoch, and use the same number of iterations in all pruning experiments for a fair comparison. When we prune the dataset, we remove the instances from each class by the same amount, so as to preserve the original proportion between classes. As will be shown in Section G, the distribution of scores is different among classes, so an imbalance problem between classes may occur if the data pruning is performed without considering the portion of classes within the dataset.

## E    ROBUSTNESS OF CG-SCORE OVER MODEL VARIANTS

Our CG-score is derived based on the theoretical analysis of the generalization error bounds on overparameterized two-layer ReLU activated neural network. To demonstrate that our CG-score is effective in more complicated networks, in the main text (Section 4.1), we used ResNets to evaluate our score for data pruning experiments. To further demonstrate the robustness of our score against model changes, in this section, we report the results of data pruning experiments on CIFAR-10 dataset using DenseNetBC-100 Huang et al. (2017), a more complicated convolutional network, and Vision Transformer (ViT) (Dosovitskiy et al., 2021) pretrained by ImageNet dataset. The implementation details of DenseNet follows that of Huang et al. (2017). We use the same hyperparameter and optimization method summarized in Table 3 for training DenseNet. To fine-tune ViT, we follow the implementation details outlined in Dosovitskiy et al. (2021). Specifically, we download a ViT model pretrained on the ImageNet dataset using the timm module in PyTorch, and then fine-tune

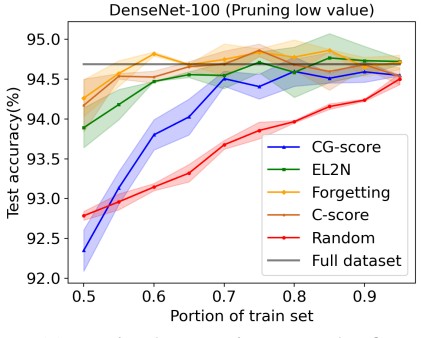
(a) Pruning low-scoring examples first.

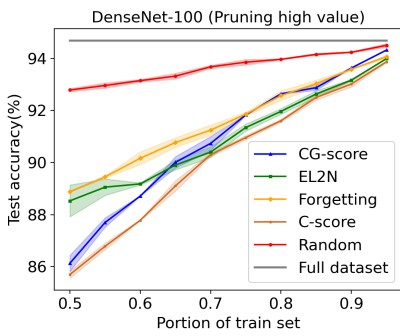
(b) Pruning high-scoring examples first.

Figure 10: Pruning experiments with CIFAR-10 dataset trained on DenseNet-100. Even if the model changes from ResNet to DenseNet, we observe similar trends as in Figure 2.

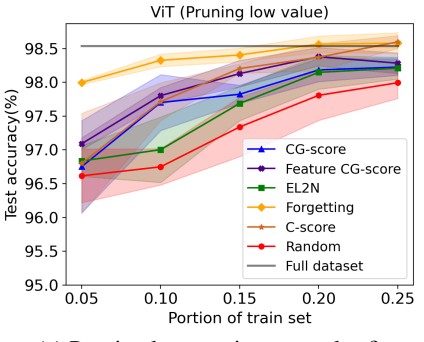
(a) Pruning low-scoring examples first.

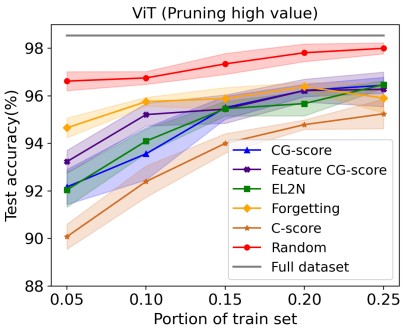
(b) Pruning high-scoring examples first.

Figure 11: Pruning experiments with CIFAR-10 dataset trained on ViT pretrained by ImageNet dataset. Despite changing the model from ResNet to ViT and fine-tuning it, we still observe similar trends as shown in Figure 2.

the model on the CIFAR-10 dataset for 10 epochs using the ADAM optimizer. We set the initial learning rate to 1e-5 and do not use a learning rate scheduler or weight decay.

In Figure 10, the left figure shows the test accuracy of the model trained with different subset sizes, when low-scoring (regular) examples are removed first, and the right figure shows the similar result but when high-scoring (irregular) examples are removed fist. We report the mean of the result after two independent runs, and the shaded regions indicate the standard deviation. The red-curve (random) is the result when randomly ordered examples are used. We can observe a similar trend as in Fig. 2, the experimental results using ResNets, in spite of the model change. Our CG-score achieves competitive performances as the other baselines. The reason that we observe a rapid performance degradation at a smaller training data portion in the left figure (removing low value data first) is due to the characteristics of leave-one-out method we used in the calculation of the CG-score. Removing a typical sample from a dataset does not change the generalization error bounds much, since similar samples already exist in the dataset. Thus, typical samples tend to have low CG-scores when the score is measured by the leave-one-out method. However, when we remove 50% of instances, samples sharing the typicality can be excluded simultaneously from the dataset, which might cause severe degradation of the generalization capability of the neural network. Thus, for a smaller training data portion, it can be better to make sure at least a small portion of typical samples is indeed included in the training. Similar observations have been made in Swayamdipta et al. (2020).

As shown in Sorscher et al. (2022), data scoring can also be effective in reducing the amount of data for fine-tuning pre-trained models. Therefore, we examine whether our CG-score can be effective for fine-tuning the transformer-based models, in particular, ViT. In Figure 11, the left figure shows the test accuracy of ViT fine-tuned with different subset sizes of CIFAR-10 when low-scoring (regular) examples are removed first, while the right figure shows the results when high-scoring (irregular) examples are removed first. We report the mean of the results after five independent runs and indicate the standard deviation with shaded regions. The red curve (random) represents the results when examples are randomly removed. We observe that when low-scoring examples are removed first, similar to Figure 2, the networks maintain test accuracy even when up to 90% of the dataset is removed. Our CG-score achieves competitive performance compared to other baselines and significantly outperforms the random baseline.

## F    ADDITIONAL EXPERIMENTS WITH TWO MORE BASELINES

### F.1    OTHER DIRECTIONS OF DATA VALUATION

There are two additional branches of related works for data valuation, in addition to the methodologies described in the Section 2. The first branch uses the influence function. The influence function approximates the degree of change of parameters when specific data enters or leaves the training dataset, so it determines which data is valuable by calculating the effect of the data on learning. The second branch uses coreset selection. Coresets are weighted subsets of the data selected to resemble the model training using the full dataset. Coresets may need to be updated as training progresses.

**Baseline algorithms for data valuation**    We use representative scores in each branch, TracIn (Pruthi et al., 2020) for influence function and CRAIG (Mirzasoleiman et al., 2020) for coreset selection, as additional baselines. Detailed definitions of the scores are described below:

- TracIn: TracIn CheckPoint (TracInCP) value between two data points is defined as the weighted sum of dot products of the loss gradients calculated at the two data points. The gradients are obtained from the $k$ checkpoints $\{t_1, \ldots, t_k\}$ of the model in the middle of training and the weight $\eta_{t_i}$ is the learning rate at each checkpoint $t_i$:

$$\text{TracInCP}(z, z') = \sum_{i=1}^{k} \eta_{t_i} \nabla l(w_{t_i}, z)^\top \nabla l(w_{t_i}, z'), \qquad (11)$$

  where $l(w_{t_i}, z)$ is loss function at $t_i$-th step with model parameter $w_{t_i}$.

- CRAIG: CoResets for Accelerating Incremental Gradient descent (CRAIG) is an algorithm that solves the optimization problem, which finds a subset that preserves the gradient of the total loss:

$$S^* \in \text{argmax}_{S \subset V} \sum_{i \in V} \min_{j \in S} \max_{w \in W} \|\nabla f_i(w) - \nabla f_j(w)\|, \text{ s.t. } |S| \le r \qquad (12)$$

  where $f_i(w) = l(w, (x_i, y_i))$ is loss for the data $(\mathbf{x}_i, y_i)$ with model parameter $w$, $V$ is the full dataset and $S$ is the coresets with size $r$, and $p_i$ be the softmax output for data $(\mathbf{x}_i, y_i)$. In CRAIG, the gradient $f_i(w)$ is approximated by $p_i - y_i$ when cross entropy loss is used with soft-max at the last layer.

### F.2    DATA PRUNING EXPERIMENTS

In this section, we conduct data pruning experiments, similar to Section 4.1. We conduct the experiments using the above two additional baselines, TracIn and CRAIG, separately from the experiments of the main text, since the two algorithms require additional assumptions/resources that have not been used for the baselines considered in the main text: TracIn requires a validation set to calculate the data values; CRAIG does not select a fixed subset of data to be used throughout the training, but keeps updating the subset of the data (coresets) to be used for training every 10 epochs.

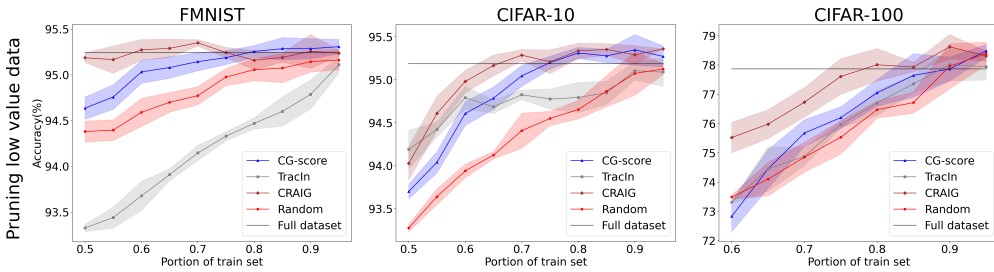

(a) Pruning low-scoring examples first. Better score maintains the test accuracy longer.

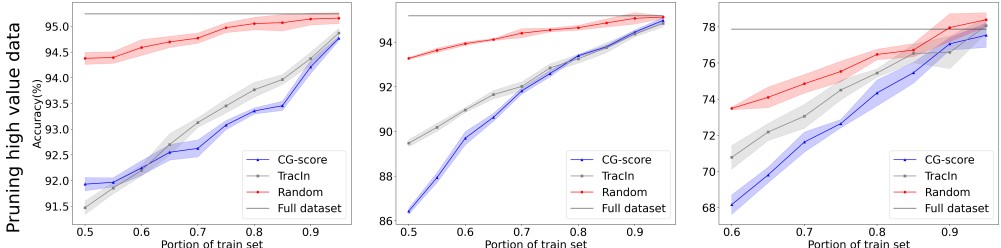

(b) Pruning high-scoring examples first. Better score makes the rapid performance drop.

Figure 12: Pruning experiments with three datasets FMNIST (left), CIFAR-10 (middle) and CIFAR-100 (right). Our CG-score can achieve better performances than the Tracin score and competitive performances compared to the CRAIG algorithm. Different from our scoring method, TracIn requires a validation set to calculate the data values and CRAIG does not select a fixed subset of data to be used throughout the training, but keeps updating the subset the data (coresets) to be used for training every 10 epochs. CRAIG has been evaluated only for pruning low-valued samples due to the nature of coreset selection where coresets are selected with per element weights.

**Experimental details**   As described in the Pruthi et al. (2020), we calculate the TracIn score by using the gradients of the parameters of the network's last layer. The check points are set at every 20 epochs, starting from the end of the first 20th epoch. We use 5 checkpoints for FMNIST and 10 checkpoints for CIFAR-10 and CIFAR-100. We create a validation set composed of 1,000 samples by taking a part of the test dataset, and calculate TracIn score with this validation set. As TracIn score is defined between two data points $(z, z')$ as in equation 11, we set the score of each training sample $z$ by averaging TracIn scores TracInCP$(z, z')$ over all samples $z'$ in the validation set.

In CRAIG, the subset selection is performed every 10 epochs. We only test CRAIG in pruning low-valued data but not in the reverse order (pruning high-valued data), since CRAIG extracts coresets to be used with per element weights for preserving the gradient of total loss but does not give what are the high-valued (equal weight) samples.

TracIn score and CRAIG are calculated at the networks same as those used in the experiment in Section 4.1: ResNet18 for FMNIST and CIFAR-10, and ResNet50 for CIFAR-100 dataset. The other experimental details are the same as Table 3 in Appendix §D.3

**Experimental results**   Similar to data pruning experiment in Section 4.1, in Figure 12, the first row shows the test accuracy of the model trained with different subset sizes, when low-scoring (regular) examples are removed first, and the second row shows the similar result but when high-scoring (irregular) examples are removed first. We report the mean of the results after five independent runs, and the shaded regions indicate the standard deviation. The red-curve (random) is the result when randomly ordered examples are used. When pruning low-scoring data first, it is preferable to maintain the accuracy up to a large removing portion (a small training set); when pruning high-scoring data first, the rapid drop of performance is preferable since it means that the score can detect high-value samples, necessary for generalization of the model. Our CG-score can achieve better performances than the Tracin score and competitive performances compared to the CRAIG algorithm. Since CRAIG updates the coresets over the training, it can choose different semantics

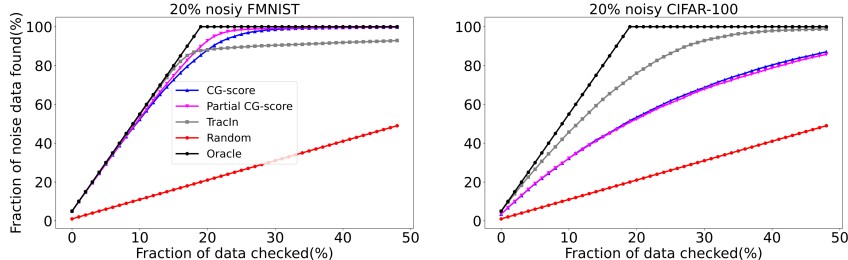

Figure 13: Fraction of label noise (y-axis) included in the examined portion (x-axis) for 20% label noise for oracle, Partial CG-score (ours), CG-score (ours), TracIn and random baseline.

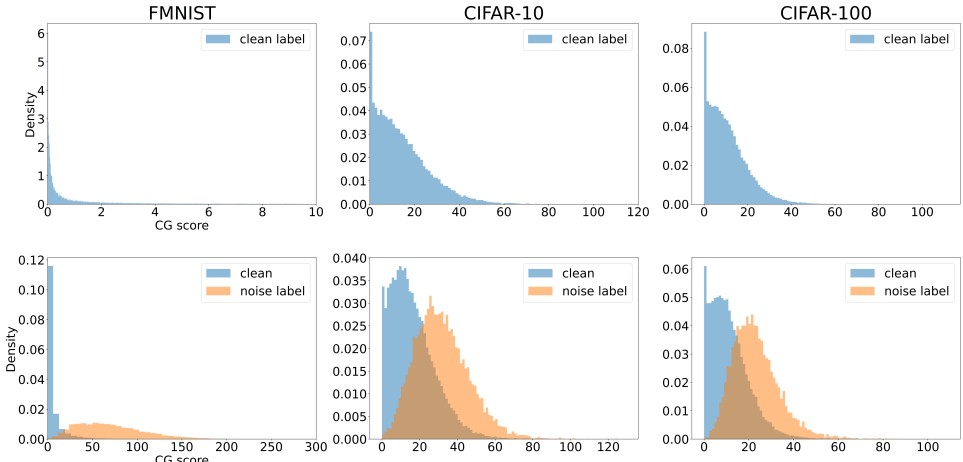

Figure 14: Density of CG-scores at clean dataset and label noisy dataset.

of the data suitable for each phase of the training, which results in better performance. This result may imply the effectiveness of the scheduled batch selection, e.g., curriculum learning, in training neural networks. We inspect that the performance of TracIn may heavily depend on the size of the validation set and also the possible domain discrepancy between training and test datasets. In our test, there is no domain discrepancy, but if there exists a domain shift between the test dataset and the training dataset, TracIn may perform better than other methods with the help of validation set.

## F.3 DETECTING LABEL NOISE

We also compared the performance of our CG-score in detecting mislabeled data with that of TracIn. As suggested in Pruthi et al. (2020), we use 'self-influence' of each training example, i.e., the influence of a training point on its own loss during the training process, TracInCP$(z, z)$, to identify mislabeled data. In Pruthi et al. (2020), it was shown that mislabeled examples tend to have higher TracIn values, and thus TracIn values can be effectively used in identifying mislabeled examples. In Fig. 13, we show the comparison of our method with TracIn in identifying mislabeled examples in FMNIST and CIFAR-100 datasets, respectively, each of which includes 20% label noise. TracIn achieves better performance in CIFAR-100, but ours outperformed TracIn in FMNIST. Since TracIn measures the 'self-influence' of each training example over the training, starting from 20th epoch, for relatively simpler dataset such as FMNIST, some mislabeled instances could have already been memorized at the network, which makes them not detectable by the TracIn values. On the other hand, our method better detects mislabeled data for easier datasets as discussed in Sec. 4.2. Thus, we can conclude that depending on data complexity, the outperforming method can be changing.

## G SCORE DISTRIBUTION FOR PUBLIC DATASETS

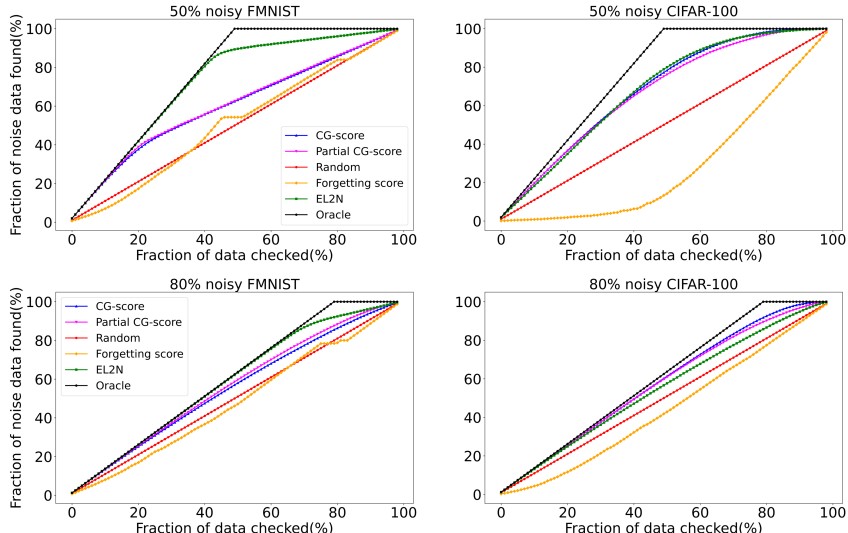

Figure 15: Fraction of label noise (y-axis) included in the examined portion (x-axis) for 50%(top) and 80%(bottom) label noise for FMINST (left) and CIFAR-100 (right) datasets.

**Score distributions** In Fig. 14, we compare the CG-score distributions of three public datasets, FMNIST, CIFAR-10 and CIFAR-100 before and after we artificially corrupt 20% of instances with random label noise. The top row shows the distributions before the corruption, and the bottom row shows the distributions after the corruption, where the orange color displays the distribution of the CG-score for 20% instances with label noise and the blue displays that for 80% instances with clean label. The gap between scores of clean data and noisy data, which enables us to detect the noise by the CG-score, is larger for relatively simpler dataset, FMINST, than those for CIFAR-10/100.

**Detecting mislabeled instances at high noise rate** We conduct additional experiments to check the detectability of mislabeled instances by our CG-score, when the noise ratio increases to 50% and even to 80% for FMNIST and CIFAR-10 dataset. In the main text, we considered a mild noise ratio of 20% and reported the result in Fig. 3. The results for higher noise cases are shown in Fig. 15. In the CIFAR-10 dataset, when the noise ratio is 80%, each class includes 1,000 correctly labeled images and 4,000 mislabeled images, composed of 445 samples coming from each of the other nine classes. Even for such an extremely noisy case, our CG-score can effectively detect the mislabeled data, since our score can discover samples that have a relatively lower correlation to the majority of the samples of the same class as analyzed in Sec. 3.3.

## H FEATURE-SPACE CG-SCORE

Our original CG-score can be calculated by data without any trained network. In this section we check the validity of a new CG-score, which is defined in terms of feature of data, and compare its characteristic with that of the original data-centric CG-score. We train ResNet18 using FMNIST and CIFAR-10 dataset and ResNet50 using CIFAR-100 dataset to obtain each data's feature at 10th epoch, where the feature is defined at the output of the penultimate layer (512 and 2048 dimensions for ResNet18 and ResNet50, respectively). Then we calculate the feature-space CG scores, which are calculated by the feature instead of the data itself. Unlike CG-score, calculating feature-space CG-score requires training of a model for a few epochs, which implies that the score becomes dependent to the model and we need additional computational efforts. However, feature CG-score achieves superior performances at data valuation as the results of data pruning experiment (Fig. 16) and noise detection experiment (Fig. 17) show even though the features were extracted at early stage of the training (10th epoch).

Furthermore, we compare how the Spearman rank correlation between the feature-space CG-scores and other scores, including the original CG-score and the previous training-based scores such as

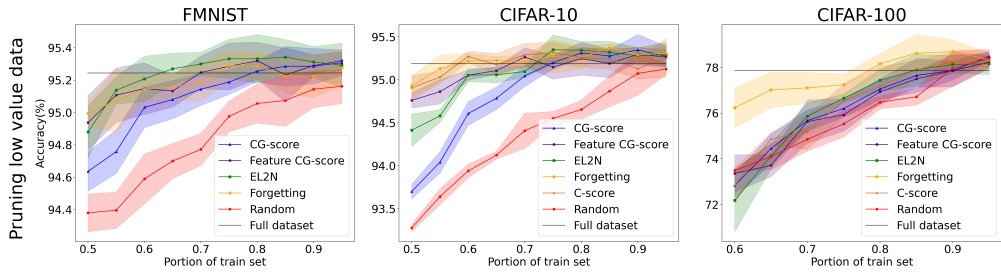

(a) Pruning low-scoring examples first. Better score maintains the test accuracy longer.

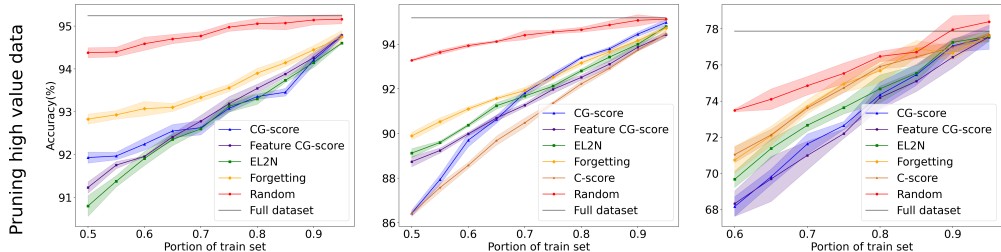

(b) Pruning high-scoring examples first. Better score makes the rapid performance drop.

Figure 16: Pruning experiments with three datasets FMNIST (left), CIFAR-10 (middle) and CIFAR-100 (right). With additional computation (training of a model for 10 epochs), our feature-space CG-score can achieve better performances than the original CG-score. The performances are competitive to other baseline scores

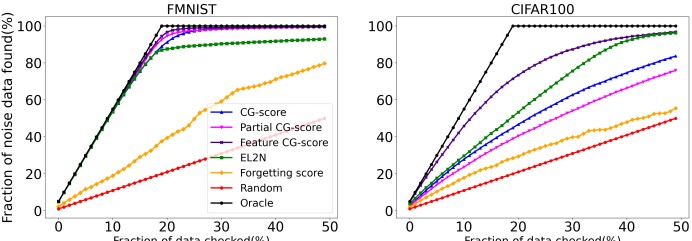

Figure 17: Fraction of label noise ($y$-axis) included in the examined portion ($x$-axis) for 20% label noise. Feature-space CG-score achieves superior noise detectability for FMNIST than for CIFAR-100.

C-score, Forgetting score, and EL2N, change over the epochs at which the feature space CG-scores are calculated. Figure 18 reports the result. In Figure 18, we can first observe that the feature space CG-score and the original CG-score has the highest correlation at the very beginning of the training (epoch 1), but as the training progresses, the correlation decreases. This trend can be explained by the fact that the feature space CG-score computes the value of data in the learned embedding space, while our score computes the value of the original data without embedding it into a latent space. As training of a model progresses, the embedded data may incur bias in the data valuation, depending on a particular model or training algorithm, and thus the correlation between the feature space CG-score and the original data-centric CG-score may decrease. On the other hand, correlation between feature space CG-score and other training-based scores (EL2N, Forgetting, and C-score) increases during the initial stage, and then decreases gradually. More specifically, for both EL2N and Forgetting score, which are computed at the same network as that of the feature space CG-score, the correlation increases as the epoch increases and then slightly drops and becomes saturated at a certain value. The C-score, which was calculated in a different CNN network, shows a slightly different tendency compared to EL2N or forgetting score, and attains its peak at an earlier epoch, epoch 5. This result implies that the feature-space CG score includes meaningful information for

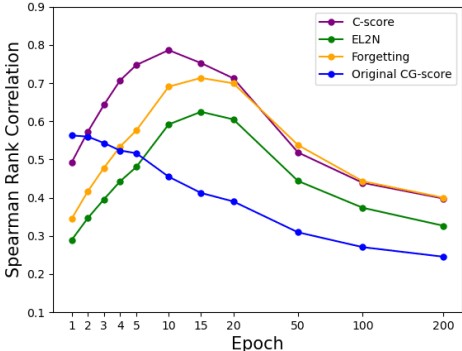

Figure 18: Spearman rank correlation (y-axis) between feature space CG-score calculated at each epoch (x-axis) and baseline scores including C-score, EL2N, Forgetting score and the original data-centric CG-score. The feature space CG-scores are calculated right after epoch 1(after one epoch of training), 3, 5, 7, 9, 11, 20, 50, and 200 (end of the training).

data valuation, correlated with other training-based scores for overall epochs. However, the feature space CG-score may incur some bias in data valuation as the training progresses. Understanding the effectiveness of the feature-space CG score in diverse applications can be an interesting future research direction.

## I   DETAILS OF KERNEL VELOCITY

We calculate the kernel velocity of 10 groups of instances, where each group is composed of 125 samples. The following is how we construct each group: Sort CIFAR-10 examples in ascending order by the CG-score, divide the examples into 10 groups, and select 125 consecutive samples from the beginning of each group.

We calculate the NTK kernel velocity as described in Paul et al. (2021): Let C be the number of classes. Let $f_t^{(c)}(\mathbf{x}_i)$ be the $c$-th logit value for input $\mathbf{x}_i$ at the $t$-epoch. When $\mathbf{W}(t)$ is the parameters of the model, the $c$-th logit gradient at input $\mathbf{x}_i$ is $\psi_t^{(c)}(\mathbf{x}_i) = \nabla_{\mathbf{W}(t)} f_t^{(c)}(x_i) \in \mathbb{R}^N$. Then, the data-dependent NTK submatrix of a group of $m$ samples $S := \{\mathbf{x}_{a_1}, \dots, \mathbf{x}_{a_m}\}$ is defined as $K_t(S) = \Psi_t(S)\Psi_t(S)^\top$, where $\Psi_t(S) \in \mathbb{R}^{mC \times N}$ is constructed by placing $\{\psi_t^{(c)}(\mathbf{x}_{a_i})\}_{c \in [C], i \in [m]}$ in rows of $\Psi_t(S)$. The kernel velocity is defined as

$$v_t(S) = 1 - \frac{\langle K_t(S), K_{t+1}(S) \rangle}{\|K_t(S)\| \, \|K_{t+1}(S)\|}. \tag{13}$$

Lastly, we provide some explanations of what the kernel velocity measures if we define it for finite-width ReLU activated 2-layer neural network. Remind that our analysis uses the Gram matrix $\mathbf{H}^\infty$ defined in equation 1, which is derived for an overparameterized ReLU activated 2-layer neural network. We can generalize the definition of the Gram matrix assuming a finite-width network, similar to $K_t(S)$, as follows. The output of the network is $f_{\mathbf{W},\mathbf{a}}(\mathbf{x}) = \frac{1}{\sqrt{m}} \sum_{r=1}^m a_r \sigma(\mathbf{w}_r^\top \mathbf{x})$, and the gradient with respect to $\mathbf{W} = (\mathbf{w}_1, \dots, \mathbf{w}_m) \in \mathbb{R}^{d \times m}$ is $\nabla_{\mathbf{W}} f(\mathbf{x}_i) = [\nabla_{\mathbf{w}_1} f(\mathbf{x}_i), \nabla_{\mathbf{w}_2} f(\mathbf{x}_i), \dots, \nabla_{\mathbf{w}_m} f(\mathbf{x}_i)] = \frac{\mathbf{x}_i^\top}{\sqrt{m}} [a_1 \mathbb{1}\{\mathbf{w}_1^\top \mathbf{x}_i \geq 0\}, a_2 \mathbb{1}\{\mathbf{w}_2^\top \mathbf{x}_i \geq 0\}, \dots, a_m \mathbb{1}\{\mathbf{w}_m^\top \mathbf{x}_i \geq 0\}]$. Denoting the network parameters at the $t$-th step as $\mathbf{W}(t)$, we can define the Gram matrix $\mathbf{H}_t$ as $(\mathbf{H}_t)_{ij} = \nabla_{\mathbf{W}(t)} f(\mathbf{x}_i) \nabla_{\mathbf{W}(t)} f(\mathbf{x}_j)^\top = \mathbf{x}_i^\top \mathbf{x}_j \frac{1}{m} \sum_{r=1}^m \mathbb{1}\{\mathbf{w}_r^\top \mathbf{x}_i \geq 0, \mathbf{w}_r^\top \mathbf{x}_j \geq 0\}$, which is proportion to the number of neurons in the hidden layer activated both for $\mathbf{x}_i$ and $\mathbf{x}_j$ at the epoch. We can define the kernel velocity similar to equation 13 by calculating $\mathbf{H}_t$ for a subset of data, and replacing $K_t(S)$ by $\mathbf{H}_t(S)$. For this case, the high kernel velocity implies that $\mathbf{H}_t$ differs much from $\mathbf{H}_{t+1}$, i.e., the portion of neurons activated for pairs of instances changes rapidly during training.

Table 5: Spearman rank correlations between the CG-score and the partial CG-scores

| Dataset | CIFAR-10 | | | | CIFAR-100 | | | |
|---|---|---|---|---|---|---|---|---|
| Subsampling Ratio | 1:1 | 1:2 | 1:3 | 1:4 | 1:1 | 1:2 | 1:3 | 1:4 |
| $d_i^{-1}(\mathbf{y}_{-i}^\top \mathbf{h}_i)^2$ | -0.759 | -0.461 | -0.136 | 0.132 | -0.671 | -0.296 | 0.035 | 0.264 |
| $2y_i(\mathbf{y}_{-i}^\top \mathbf{h}_i)$ | 0.912 | 0.948 | 0.962 | 0.970 | 0.902 | 0.939 | 0.955 | 0.964 |
| $y_i^2 d_i$ | -0.015 | 0.066 | 0.121 | 0.163 | -0.069 | 0.018 | 0.087 | 0.136 |

## J    CORRELATIONS BETWEEN CG-SCORE AND PARTIAL CG-SCORES

From the CG-score, which is expressed as $d_i^{-1}(\mathbf{y}_{-i}^\top \mathbf{h}_i)^2 + 2y_i(\mathbf{y}_{-i}^\top \mathbf{h}_i) + y_i^2 d_i$ at the equation 6, we can define three partial CG-scores, $d_i^{-1}(\mathbf{y}_{-i}^\top \mathbf{h}_i)^2$, $2y_i(\mathbf{y}_{-i}^\top \mathbf{h}_i)$, and $y_i^2 d_i$, respectively. In this section, we analyze which term dominates the order of CG-scores for two public datasets, CIFAR-10/100, by analyzing the correlation between the CG-score and the three partial CG-scores. We also examine whether the correlations vary when we change the size of the Gram matrix $\mathbf{H}^\infty$, whose dimension changes according to the level of sub-sampling, explained in Sec. A.2. We get $\mathbf{H}^\infty$ with different subsampling ratios, 1:1, 1:2, 1:3, and 1:4, and then calculate the CG-score and partial CG-scores, defined by the three partial terms. Table 5 shows the Spearman rank correlations between the CG-scores and the partial CG-scores. We can see that $2y_i(\mathbf{y}_{-i}^\top \mathbf{h}_i)$, which was utilized at noise detection experiments, is the partial score having the largest correlation with the CG-score across all subsampling ratios. On the other hand, correlations to other partial-scores are relatively low, and the correlations change much as the subsampling ratio changes.

## K    SCHUR COMPLEMENT

Calculating CG-score (equation 2) of $n$ data instances requires the inversion of $n \times n$ matrix, which may cause expensive computational cost for a large $n$. Therefore, we provided computationally efficient way to obtain the CG-score by using Schur complement equation 4. Here we provide some details to derive the inverse of a sub-block matrix using Schur complement. By Schur complement, we have

$$\begin{pmatrix} \mathbf{A} & \mathbf{B} \\ \mathbf{C} & \mathbf{D} \end{pmatrix}^{-1} = \begin{pmatrix} (\mathbf{A} - \mathbf{B}\mathbf{D}^{-1}\mathbf{C})^{-1} & -(\mathbf{A} - \mathbf{B}\mathbf{D}^{-1}\mathbf{C})^{-1}\mathbf{B}\mathbf{D}^{-1} \\ -\mathbf{D}^{-1}\mathbf{C}(\mathbf{A} - \mathbf{B}\mathbf{D}^{-1}\mathbf{C})^{-1} & \mathbf{D}^{-1} + \mathbf{D}^{-1}\mathbf{C}(\mathbf{A} - \mathbf{B}\mathbf{D}^{-1}\mathbf{C})^{-1}\mathbf{B}\mathbf{D}^{-1} \end{pmatrix}. \quad (14)$$

Remind that $\mathbf{H}^\infty$ and $(\mathbf{H}^\infty)^{-1}$ are denoted by

$$\mathbf{H}^\infty = \begin{pmatrix} \mathbf{H}_{n-1}^\infty & \mathbf{g}_i \\ \mathbf{g}_i^\top & c_i \end{pmatrix}, \quad (\mathbf{H}^\infty)^{-1} = \begin{pmatrix} (\mathbf{H}^\infty)_{n-1}^{-1} & \mathbf{h}_i \\ \mathbf{h}_i^\top & d_i \end{pmatrix}, \quad (15)$$

where $\mathbf{H}_{n-1}^\infty, (\mathbf{H}^\infty)_{n-1}^{-1} \in \mathbb{R}^{(n-1)\times(n-1)}$, $\mathbf{g}_i, \mathbf{h}_i \in \mathbb{R}^{n-1}$ and $c_i, d_i \in \mathbb{R}$, i.e.,

$$(\mathbf{H}^\infty) = \begin{pmatrix} (\mathbf{H}^\infty)_{n-1}^{-1} & \mathbf{h}_i \\ \mathbf{h}_i^\top & d_i \end{pmatrix}^{-1} = \begin{pmatrix} \mathbf{H}_{n-1}^\infty & \mathbf{g}_i \\ \mathbf{g}_i^\top & c_i \end{pmatrix} \quad (16)$$

By substituting $\mathbf{A}$, $\mathbf{B}$, $\mathbf{C}$, and $\mathbf{D}$ in equation 14 with $(\mathbf{H}^\infty)_{n-1}^{-1}$, $\mathbf{h}_i$, $\mathbf{h}_i^\top$, and $d_i$ respectively, we obtain that

$$\mathbf{H}_{n-1}^\infty = ((\mathbf{H}^\infty)_{n-1}^{-1} - \mathbf{h}_i\mathbf{h}_i^\top/d_i)^{-1}. \quad (17)$$

Thus, $(\mathbf{H}_{n-1}^\infty)^{-1} = (\mathbf{H}^\infty)_{n-1}^{-1} - \mathbf{h}_i\mathbf{h}_i^\top/d_i$.

