# OpenReview forum: "Data Valuation Without Training of a Model"
_ICLR.cc/2023/Conference — ICLR 2023 poster_

### Official Review · Reviewer_3m6j · 2022-10-20

**Confidence:** 3
**Correctness:** 2
**Technical Novelty And Significance:** 3
**Empirical Novelty And Significance:** 2
**Recommendation:** 3

**Clarity, Quality, Novelty And Reproducibility:**

The writing quality was fine overall.  At points, the writing was, in places, quite wordy with sentences too long (e.g., the first sentence in the abstract).  Similarly, paragraphs were often too long.  These long sentences/paragraphs detract from the paper's clarity and from the ability of the reader to grasp the authors' main point.  I did not base my score on this point much, and this is primarily provided as feedback to the authors.

The authors do not provide an implementation of their method.  The experiments are simple enough many of them could be implemented.
* Figure 5's caption lacks many of the key experimental details. The description of the experiment in Sec. A.3 is not clear or easy to follow.  Also, the number of trials in each experiment is not clear.

A non-exhaustive list of typos:
* Pg. 2: 'has a close relation to 'learning *difficult*' of the instances'
* Pg. 8: 'by analyzing the training dynamics *of CIFAR10 dataset* in'
* Pg. 13: '*filpping*'

**Details Of Ethics Concerns:**

None.

**Strength And Weaknesses:**

Reading the paper's abstract, I was quite excited by the stated claims.  The ideas and the paper have promise.  However, the paper has problems; it is not yet ready for publication and needs to empirically evaluate and candidly discuss where this "training-free" data valuation runs into limitations.

#### **Strengths**

- A competent method to evaluate the quality of data without model training is clearly highly valuable.  This makes CG score's motivation clear and compelling.

- The generalization of complexity score from a dataset valuation method to an instance valuation method is a natural strategy.  It makes the paper's theoretical foundation easier to understand.

- I appreciated the discussion of CG', and I am glad the authors included it.

#### **Weaknesses**
- Evaluating only Fashion MNIST and CIFAR10(0) is very disappointing.  These are not particularly challenging datasets.  Related work (e.g., Jiang et al. 2020) evaluate on ImageNet.  I could not recommend acceptance without more challenging datasets.

- I am surprised about Table 4's poor rank correlation between "feature space CG" and vanilla CG.  It is concerning how well your method will work on more challenging tasks.  Moreover, I imagine that pretrained features will generally be more semantically meaningful than the raw pixel values.  My intuition could be wrong here, but I imagine I would not be the only one to hold such a prior.  The authors really need to address this point at least in Section G but ideally in the main paper.

- In Section 3.1, the authors describe CG score as a "*data-centric measure*."  However, all evaluation in this paper is on vision datasets.  Data-centric/data-agnostic measures should be evaluated on multiple data modalities.  At best, this evaluation could only show that CG score is an effective vision data metric.

- Figure 2 was confusing at first review.  I believe a primary reason for that is that in one case "larger is better" and in the other "smaller is better."  Perhaps dividing it into two subfigures where that difference is made more obvious (e.g., in the caption) would improve clarity.

- The paper as a whole needs more baselines.  There are many alternate data valuation approaches (e.g., Influence Functions, TracIn, Shapley Value methods) not evaluated here that should be for at least some experiments
  - Figure 3b should at least include C-score.  I understand Jiang et al. do not provide precalculated C-score values for FashionMNIST but they do for standard MNIST.  Hence, Figure 3b's evaluation could use regular MNIST. Jiang et al. also provide C-score for CIFAR100.

- This paper needs a self-contained section explicitly specifying the computational and space complexity. It is fine to include it in the supplement, but it needs to be there.  If I missed it, please let me know, but I also checked the supplement.  Section A.2 came the closest but largely glossed over the point.
  - I assume you have data on the execution time of your method.  It should be in the supplement for reference.

- The "*Related Works*" does not adequately discuss the relation between data valuation and training-set influence. This relationship is primarily contextualized through Data Shapley which is only one method in this broader field. I also think the evaluation would be stronger with one or more training-set influence methods (e.g., TracIn [Pruthi et al. 2020]) included.

**Summary Of The Paper:**

This paper proposes the complexity gap (CG) score, which quantifies the change in data complexity when each training instance is removed from the training set.  The primary differentiator between CG score and other data valuation measures (e.g., Shapley value, C-score, etc.) is that CG score can be calculated without any (partial) model training.

**Summary Of The Review:**

The concept of estimating data's value without any training is obviously attractive. However, there are no free lunches. "Low-cost" strategies usually require tradeoffs and encounter limitations where the method breaks down.  The authors never really address this point by, for example, affirmatively showing that such a "breakdown" does not occur on hard tasks. They also do not include a substantive, stand-alone discussion or acknowledgment of the method's observed "breaking points."  That is a grievous omission.

If the authors have evaluated their method on harder tasks and did not report the results, those results are needed.  Without them, the paper is definitely incomplete.

The paper has promise. Perhaps with a strong author response, I could be persuaded to acceptance. However, the paper changes, in particular the additional necessary evaluation, needs significant revision.

---

> ### Author Response · Authors · 2022-11-18
> **Response to Reviewer 3m6j (3/3)**
>
> >**7. The paper as a whole needs more baselines. There are many alternate data valuation approaches (e.g., Influence Functions, TracIn, Shapley Value methods) not evaluated here that should be for at least some experiments.**
>
> As the reviewer suggested, we evaluated two more recent baselines, CRAIG [b] and TracIn [c], each of which represents two different approaches for data valuation 1) coresets selection and 2) influence function, respectively, and reported the results in Appendix D.
>
> CRAIG [a] provides generic algorithms to select a weighted representative subset that closely approximate the full gradient and achieves the one of the best performances in training NNs among coreset selection schemes. Thus, we used CRAIG as a representative baseline for coresets. However, we'd like to emphasize that CRAIG requires the training of a model to find the coresets that approximate the full gradient. Moreover, CRAIG keeps updating the coresets over the training, and thus it may not be a suitable algorithm for dataset pruning applications, where we want to select a “fixed” subset of data (possibly at the beginning of the training) that will be used throughout the training.  Nonetheless, we compared the performance of CRAIG in the dataset pruning experiment with our method in Figure 7-(a) of Appendix D.2 to examine the effectiveness of updated coresets over the training. The coresets selected from CRAIG are updated every 10 epochs, a total of 20 times for 200 epochs of training.  With the updated coresets, CRAIG achieved a better performance than our method in the pruning experiment. This result may imply the effectiveness of the scheduled batch selection, e.g., curriculum learning, in training neural networks.
>
> As the reviewer suggested, we also compared the performance of our CG-score with TracIn [c]. To calculate the TracIn scores, we need an additional validation set, which has not been used for other valuation methods, including our CG-score, EL2N and forgetting score. For the CIFAR-10 dataset, we designed the validation set composed of 1,000 samples randomly selected from the test set. As shown in Fig. 7, we could observe that our scoring method outperforms the TracIn both for the case of removing low-valued samples first and high-valued samples first. We inspect that the performance of TracIn might heavily depend on the size of the validation set and also the possible domain discrepancy between training and test datasets.
>
>
>
> [b] Mirzasoleiman et al., Coresets for Data-efficient Training of Machine Learning Models, ICML 2020.
>
> [c] Pruthi et al., Estimating Training Data Influence by Tracing Gradient Descent, NeurIPS 2020.
>
>
>
>
>
> >**8. This paper needs a self-contained section explicitly specifying the computational and space complexity.**
>
>
> In Table 4 of Appendix C.2, we compared the computation time (in seconds) of our CG-score with other methods. The computation time of our method depends on the dimension of the matrix we take the inversion. For CIFAR-100 dataset, each class includes 500 instances and when we calculate the CG-score by 1:4 sampling ratio between the class of interest and the rest of the classes, we need to calculate the inversion of 2,500 $\times$ 2,500 matrices. This calculation can be done within less than 1 minute, which is 1/10x  compared to the training time of ResNet18 for 20 epochs to get the EL2N score. The forgetting score, which requires the full training until 200 epochs, requires 100x times compared to the computation time of our method. For CIFAR-10 dataset, on the other hand, where each class includes 5,000 instances, with the same 1:4 sampling ratio, we need to take the inversion of 25,000 $\times$ 25,000 matrix, which requires 1.2 hours to calculate the score. This time is comparable to the training time of ResNet18 for 200 epochs (the time to calculate the Forgetting score). All the baselines as well as our method require stochastic calculation, averaging the score over multiple runs. For a fair comparison, we compared the time to get each score for a single run.
>
> >**9. The "Related Works" does not adequately discuss the relation between data valuation and training-set influence. I also think the evaluation would be stronger with one or more training-set influence methods (e.g., TracIn [Pruthi et al. 2020]) included.**
>
> We updated the related works accordingly and included TracIn as a baseline in the pruning experiment reported in Fig. 7 of Appendix D.
>
> >**10. More experimental details are required: Figure 5's caption lacks many of the key experimental details. The description of the experiment in Sec. A.3 is not clear or easy to follow. Also, the number of trials in each experiment is not clear.**
>
>
> We modified the caption of Figure 5 to include experimental details. Also, we provided more details in Appendix A.3.

---

> > ### Comment · Reviewer_3m6j · 2022-12-02
> > **Final Thoughts**
> >
> > This paper definitely has strengths.
> >
> > After the reviewer discussion, I remain of the view that this paper is not yet ready for publication at a top-tier venue.  Papers that make strong claims but then under-evaluate those claims can be a step backward for the community.  Authors are responsible for properly contextualizing their method, including any major limitations they know of.  A method having (major) limitations should not, in and of itself, be a reason for rejection.  All methods have limitations.  However, those limitations need to be made clear so a method's strengths and weaknesses can be properly weighted.
> >
> > While I appreciate the new Tiny ImageNet experiments in the updated submission, Tiny ImageNet is still relatively low resolution (64 x 64 pixels).  Tiny ImageNet is much closer to CIFAR (32x32) than full ImageNet (224x224).  I do not expect the ideas in this paper generalize to full ImageNet -- although that is just speculation since we have no experiments one way or the other.  Moreover, this paper's evaluation is on vision datasets, despite, at minimum, the paper's title positions the work as general data valuation.  From the evaluation provided, this paper provides insight into "Vision Data Valuation" on simpler datasets.
> >
> > The authors had the opportunity to better contextualize the method's limitations in their rebuttal.  It is my view that this did not occur here.  This makes me lack confidence that if this paper is accepted that those changes will be properly incorporated into the final version.

---

> > > ### Author Response · Authors · 2022-12-02
> > > **Thank Reviewer 3m6j for the reply.**
> > >
> > > Thank you for the feedback. In the revision, we have made efforts on providing additional experiments and clarifications, and we believe that we have demonstrated both the benefits and limitations of our method throughout this process. As an example, in Appendix C.2 by reporting the computational cost of our method and comparing it with those of other methods, we explained when our method can be computationally much more efficient than other training-based methods and when the benefit decreases. In Appendix D and E, by additionally comparing our method with TraIn and CRAIG in the data pruning experiment, we also explained where our method can be positioned compared to other baselines using the coresets or validation sets, respectively, and we mentioned when these other methods can achieve better performance than ours. In Appendix J, we also added why the correlation between our score and the feature space CG-score becomes lower than other training-based methods as the model training progresses.
> > >
> > > We have added all these changes and discussions mainly in the appendix during rebuttal (due to the space limitation of the main document), but we will incorporate the main discussions on the limitations of our work in the main document as well in the final version.

---

> ### Author Response · Authors · 2022-11-18
> **Response to Reviewer 3m6j (2/3)**
>
> >**2. Evaluation of the CG-score on a more challenging dataset is required. Related work (e.g., Jiang et al. 2020) evaluated on ImageNet.**
>
> As the reviewer suggested, we computed the CG-score on the high-resolution tiny ImageNet dataset (64 by 64 resolution, 100,000 samples of 200 classes). We reported the results of pruning experiments in Table R1 above and showed that our score achieves better performance than the random sampling or EL2N. However, (tiny) ImageNet is a complicated dataset and thus it was not proper to evaluate the effectiveness of valuation scores by measuring how the test accuracy maintains after pruning a significant portion of the dataset using a scoring method. Thus, as in [a], we instead used the computed CG-scores to examine how much semantics are captured by the score. We analyzed the Tiny ImageNet dataset by the CG-score distributions over 200 classes and reported the results in Figure 12 of Appendix G. We examined two easiest classes, Sulfur butterfly and Jellyfish, having relatively smaller mean and std of CG-score, and two difficult classes, Pill bottle and Syringe, having higher mean and std of CG-score. For the two easy classes, low-scoring examples look very much similar to each other and share some typical attributes (color and shape). On the other hand, for two difficult classes, even low-scoring examples do not look very similar to each other but rather diverse. From the analysis, we could check that our CG-score is still effective in examining high-resolution complicated dataset, and the score reflects the instance-wise structural regularities, which can be used in analyzing or improving learning algorithms.
>
> >**3. Table 4's poor rank correlation between "feature space CG" and vanilla CG**
>
> Thank the reviewer for the insightful comment. The reason could be that the feature space CG-score computes the value of data in the learned embedding space, while our score computes the value of the original data without embedding it into a latent space. As training of a model progresses, the embedded data may incur bias in the data valuation, depending on a particular model or training algorithm. To further examine how the correlation between “feature space CG” and original CG-score as well as other valuation scores (C-score, Forgetting score, and EL2N) change over the training, we compared the Spearman rank correlation between the feature space CG-scores calculated at different epochs with other scores. The result is reported in Fig. 15. At the very beginning of the training (epoch 1), the feature space CG-score and the original CG-score have the highest correlation since the embedding space does not incur bias yet, but as the training progresses, the correlation decreases. On the other hand, for both EL2N and Forgetting score, which are computed at the same network as that of the feature space CG-score, the correlation increases as the epoch increases and then slightly drops and becomes saturated at a certain value. The C-score, which was calculated in a different CNN network, shows a slightly different tendency compared to EL2N or forgetting score, and attains its peak at earlier epoch. From the plots, we could conclude that the feature-space CG score includes meaningful information for data valuation, correlated with other training-based scores for overall epochs. However, at the same time, the feature space CG-score may incur some bias in data valuation as the training progresses. Understanding the effectiveness of the feature-space CG score in diverse applications can be an interesting future research direction. We added this discussion in Appendix J.
>
> >**4. Data-centric/data-agnostic measures should be evaluated on multiple data modalities.**
>
> We could not add additional experiments on multiple data modalities due to time limitation, but we will keep this direction as a future work.
>
>
> >**5. Figure 2 was confusing at first review. Perhaps dividing it into two subfigures where that difference is made more obvious (e.g., in the caption) would improve clarity.**
>
> Thank the reviewer for the suggestion. We updated the figure accordingly.
>
> >**6. Figure 3b should at least include C-score. I understand Jiang et al. do not provide precalculated C-score values for FashionMNIST but they do for standard MNIST. Hence, Figure 3b's evaluation could use regular MNIST. Jiang et al. also provide C-score for CIFAR100.**
>
>
> In [a], C-scores are calculated for ‘clean’ CIFAR-10/100 datasets but not for noisy datasets. Since the scores for data instances will change by a large amount when mislabeled examples are included in the dataset, we cannot directly use the pre-computed C-scores for the noisy dataset. Since calculating C-score requires re-training of a neural network for a large number of times, we did not additionally compute the score for comparison in the experiment.

---

> ### Author Response · Authors · 2022-11-18
> **Response to Reviewer 3m6j (1/3)**
>
> We sincerely thank the reviewer for the constructive feedback. We tried our best to address the reviewer’s concerns and questions. Hope this response answers the reviewer’s questions.
>
> >**1. The concept of estimating data's value without any training is obviously attractive. However, there are no free lunches. "Low-cost" strategies usually require tradeoffs and encounter limitations where the method breaks down. The authors need address this point by, for example, affirmatively showing that such a "breakdown" does not occur on hard tasks.**
>
>
> We agree with the reviewer that there could exist scenarios where our “training-free” data valuation score, CG-score, encounters limitations. We first discuss two possible scenarios, and then report additional experiments we conducted to check the robustness of our score.
> First, since our score is developed based on the theoretical analysis of the generalization error bounds on overparameterized two-layer ReLU activated neural network, the score might be less effective compared to other model-specific scores such as EL2N, forgetting score or TracIn, when the scores are evaluated in much complicated/general neural network models. Second, as the reviewer pointed out, for more complex high-resolution datasets, such as ImageNet, the CG-score, defined based on the raw-pixel information, might not capture important semantics required for data valuation.
>
> To check the robustness of our CG-score against complicated models/datasets, we conducted additional experiments to evaluate our score in a more complicated neural network, DenseNet (Dense Convolutional Network), and on a more complicated dataset, Tiny ImageNet (64 by 64 resolutions, 100,000 images of 200 classes).
>
> Figure. 9 of Appendix E shows the results of data pruning experiment for CIFAR-10 dataset using the DenseNet. We can observe that our score is still competitive compared to other baselines, EL2N and Forgetting score, calculated using the training dynamics of DenseNet itself, both in identifying low-value samples and high-value samples. This result shows the robustness of our score against model variants.
>
> We also conducted data pruning experiments for high-resolution Tiny ImageNet dataset and compared the performances of our score with EL2N and Forgetting score as well as the random sampling (C-score could not be compared since C-score was evaluated on ImageNet but not on Tiny ImageNet, and the scores of each instance change if the dataset changes). Table below summarizes the result. We could first observe that pruning a large portion of a dataset for complicated datasets, such as Tiny ImageNet, is not proper in maintaining the performance for every baseline. For FMNIST, CIFAR-10 and CIFAR-100, when we pruned 40\% of training examples with random sampling, the test accuracy was dropped by <1\%, 2\% and 4\%. But for the Tiny ImageNet, with 40\% of data pruning, the training accuracy was dropped by almost 9\%, even though the Tiny ImageNet is the largest dataset. Our score achieved better performance than the random sampling and EL2N but worse performance than the Forgetting score.
>
>
> Table R1. Pruning experiment for Tiny ImageNet. Test accuracy(%) with varying portions of the train set when low-valued samples are removed first.
> | Data Portion (%) |  60   |  65   |  70   |  75   |  80   |  85   |  90   |  95   |
> |:----------------:|:-----:|:-----:|:-----:|:-----:|:-----:|:-----:|:-----:|:-----:|
> |     CG-score     | 54.74 | 56.43 | 58.67 | 59.48 | 59.94 | 61.02 | 61.47 | 61.96 |
> |       EL2N       | 54.38 | 55.82 | 57.23 | 58.62 | 59.71 | 61.57 | 61.67 | 62.16 |
> |    Forgetting    | 58.94 | 59.24 | 60.37 | 60.86 | 61.94 | 62.36 | 62.19 | 62.47 |
> |      Random      | 53.34 | 55.29 | 57.03 | 57.7  | 59.23 | 60.19 | 61.19 | 62.10 |
>
>
> Up to our knowledge, ImageNet has not been tested for data pruning experiments (with large pruning portions) in all the previous literatures where valuation scores are proposed. In [a], the C-score is evaluated for ImageNet to analyze the dataset, i.e., irregularity of samples or distribution of the scores over different classes, but not for data valuation or pruning. We argue that the reason could be for complicated high-resolution datasets such as ImageNet, networks often struggle in achieving good generalization capability even with the full dataset, and thus pruning a large portion of training examples may not be proper for complicated datasets.
>
> [a] Jiang et al., Characterizing Structural Regularities of Labeled Data in Overparameterized Models, ICML 2021.

---

### Official Review · Reviewer_egMu · 2022-10-23

**Confidence:** 4
**Correctness:** 4
**Technical Novelty And Significance:** 3
**Empirical Novelty And Significance:** 3
**Recommendation:** 8

**Clarity, Quality, Novelty And Reproducibility:**

The work is well written and it has decently high novelty (although it's based on ideas from Arora et al, 2019). I have no concerns about reproducibility.

**Strength And Weaknesses:**

--- Strengths ---

- A reasonable and relatively simple approach to quantify the value of individual training samples: the difference in a data complexity measure (the complexity gap score).
- An interesting application of the generalization gap analysis for two-layer networks from Arora et al (2019).
- The authors consider how to minimize computation time by reducing the number of matrix inversions.
- One of very few methods that enables dataset pruning without performing a full training run.
- The authors elaborate on the mathematical relationship between the complexity gap score several existing scores ("Correlation to other scores").
- Provides similar or better performance to existing methods in the experiments with FashionMNIST, CIFAR-10/100

--- Weaknesses ---

- Heavily based on ideas from Arora et al (2019). I personally don't view this as a major shortcoming, I think it's a valuable insight that work from a very different part of the field is applicable here.
- The authors could consider adding an appendix section to walk through their derivations for eq. 3-4 in more detail. It took me a while to see where these results come from, particularly the part about the Schur complement (it seems related to the formula for inversion of block matrices, which was not mentioned).
- Could the authors provide more information about the running-time they observe in practice? It seems important to know how the large matrix inversion $(\mathbf{H}^\infty)^{-1}$ compares to a single training run, or how much the computation is reduced when performing the stochastic calculation.
- In the experiments, the C-score doesn't appear to have been used for Fashion-MNIST (Figure 2). Is there a reason for this, would the authors be able to add it?
- One shortcoming in this approach seems to be that we expect large complexity gap scores both for useful/difficult samples and for samples with incorrect/noisy labels. This means that if we prune the dataset by keeping only samples with large scores, we're likely to retain samples with incorrect labels. I suspect this issue does not exist for any previous method that explicitly evaluates performance on a validation/test set when training with partial training sets. Could this be discussed/acknowledged somewhere in the paper?

About the method:
- It would be nice to allude to several implementation details in the main text, such as how you ensure scalar labels, $|y| < 1$, vector-shaped inputs and $||x||_2 = 1$. I see that these were described in Appendix A, but this would have been nice to point to more clearly, perhaps in the experiments section.
- The stochastic calculation approach was not mentioned at all in the main text, but it seems to play a crucial role in reducing the practical computation time. Is that what was used throughout the experiments? The sanity check with Spearman's rank correlation is helpful, but could the authors also include a plot with Pearson's correlation? I would hope that we see the results converging to something closer to 1, because Pearson's will be less sensitive to small disagreements between the lowest scores.
- Do the authors have any practical recommendations for determining the required number of runs required for convergence of their stochastic approach? Could it be significantly different for different datasets?

A couple missing works that should probably be mentioned:
- TracIn [1] belongs to the family of methods that analyze training dynamics to understand the value of each sample in the dataset. This should probably be cited and could even be considered as an additional baseline for the complexity gap score.
- There's a paper that attempts to derive a closed-form solution for the Data Shapley values [2], and I believe it doesn't require training a model either. The results end up being bounds rather than the exact values, but I think this work ends up falling in the same category. This work should definitely be mentioned, and perhaps compared against in the experiments.

[1] Pruthi et al, "Estimating training data influence by tracing gradient descent" (2020)
[2] Kwon et al, "Efficient computation and analysis of distributional Shapley values" (2021)

**Summary Of The Paper:**

This work proposes a method for scoring the utility of individual samples in a dataset, a problem that's relevant for dataset pruning and identifying incorrect/noisy labels. Whereas previous work has developed solutions that involve training the model with the full dataset one or more times, this work proposes a model-free approach. The approach focuses on the data complexity score defined by Arora et al (2019) and calculates the difference when the sample is included vs excluded. The computational complexity for this approach is potentially high, requiring the inversion of a $n \times n$ matrix (where $n$ is the training set size), but this will in some cases compare favorably to the cost of model training.

The authors provide experiments with three image datasets showing that the method (complexity gap score) is useful for dataset pruning, and that it can highlight cases where the dataset has noisy labels.

**Summary Of The Review:**

I think this work is a valuable addition to the data valuation literature. The method has some shortcomings and the paper could be improved in a couple places, but I view it favorably overall. If the authors could make some of the changes mentioned in my review, I would be inclined to raise my score.

---

> ### Author Response · Authors · 2022-11-18
> **Response to Reviewer egMu (3/3)**
>
> >**6. The stochastic calculation approach was not mentioned at all in the main text, but it seems to play a crucial role in reducing the practical computation time. Is that what was used throughout the experiments? Could the authors also include a plot with Pearson's correlation?**
>
> Yes, the CG-score used in all the experiments are calculated using the stochastic method described in Appendix A. We added a sentence in footnote 1 to explain that the stochastic method was used for the calculation of the score. We also included a plot for Pearson’s correlation in Figure 5 (right) of Appendix A. We checked that both Spearman's rank correlation and Pearson’s correlation increase as the number of runs increases, and both converge to the values close to 1, 0.973 and 0.967, respectively, at the ratio of sampling ratio of 1:4.
>
>
> >**7. Do the authors have any practical recommendations for determining the required number of runs required for convergence of their stochastic approach? Could it be significantly different for different datasets?**
>
> The proper number of runs in stochastic calculation of the CG-score may need to be determined by the size and the number of classes of the datasets. We recommend the number of runs to include at least half of the whole dataset in calculation of the CG-score for each class.  As an example, for the CIFAR-100 dataset, where each class includes 500 images, when the sampling ratio between the class of interest and the rest of classes is 1:4, we calculate the score for 500 images from the class of interest by using 2,000 images from the rest of 99 classes. Then, about 20 images are selected from each of the 99 classes. To cover at least a half of the images per class (250), we need to repeat the runs about 10 times (ignoring overlap of samples in each run). For the FMNIST/CIFAR-10 datasets, the similar calculation shows that only 2-3 runs will be enough. We added this detail in Appendix A.
>
> >**8. Missing works: TracIn and a closed-form solution for the Data Shapley values. TracIn could even be considered as a baseline.**
>
> Thank the reviewer for the suggestion. We added the two papers in the related work, and also compared our method with TracIn [b] in an additional experiment reported in Fig. 7 of Appendix D. We did not add [c] as a baseline, since the paper provides the D-Shapley, only for simple learning models (linear regression, binary classification, and density estimation) but not for multi-class classifications. To calculate the TracIn scores, we need an additional validation set, which has not been used for other valuation methods, including our CG-score, EL2N and forgetting score. For the CIFAR-10 dataset, we designed the validation set composed of 1,000 samples randomly selected from the test set. As shown in Fig. 7, we could observe that our scoring method outperforms the TracIn both for the case of removing low-valued samples first and high-valued samples first. We inspect that the performance of TracIn might heavily depend on the size of the validation set and also the possible domain discrepancy between training and test datasets, i.e., if there exists a domain shift between the test dataset and the training dataset, TracIn might perform better than ours since ours do not use any validation set for scoring training instances.
>
> [b] Pruthi et al., Estimating training data influence by tracing gradient descent, NeurIPS 2020.
>
> [c] Kwon et al., Efficient computation and analysis of distributional Shapley values, AISTATS 2021.

---

> > ### Comment · Reviewer_egMu · 2022-12-02
> > **Thanks for response**
> >
> > Thanks to the authors for their detailed response. I'm mostly satisfied with the clarifications and additions to the paper, so I'll raise my score. But I do have a couple lingering comments:
> >
> > - Regarding the stochastic calculation, mentioning this in a footnote is an improvement. But since subsampling is apparently necessary to use the method, shouldn't it be discussed more prominently? Relegating it to a footnote makes it seem optional and increases the likelihood readers will miss it.
> >
> > - Regarding large CG scores for unusual (but helpful) and mislabeled data, thanks for highlighting results where very large scores for mislabeled examples enable separating them. This is helpful, but I don't think it's accurate to say that this issue is universal among data valuation methods. Take Data Shapley, for example: in this method, unusual but helpful examples improve accuracy and get large scores, while mislabeled examples hurt accuracy and get low scores. Thus, the key difference seems to be between 1) methods that directly test impact on performance (TracIn falls in this category too) and 2) methods that test (in some form) irregularity wrt the data distribution. This method appears to fall into category 2), although the specific way that irregularity is tested is complex. Anyway, this difference should probably be highlighted, and in my opinion the unusual/mislabeled example issue should be acknowledged as well.
> >
> > - It would be nice to test on ImageNet, but since it's not common in this area and requires large computational resources, not including it seems fine. However, perhaps you could comment on how you expect computation to scale to ImageNet-size datasets though. For example, do you expect you would need to increase the number of subsampling rounds substantially, or the size of the subsampled datasets?

---

> > > ### Author Response · Authors · 2022-12-02
> > > **Thank Reviewer egMu for the reply.**
> > >
> > > Thank you for the positive comments and the suggestion to better explain our work. We will modify it in the final version. Additionally, we hope that the additional responses resolve your concerns.
> > >
> > > * We agree with the reviewer that it’d better to introduce the stochastic calculation in the main document rather than in the footnote. We will modify it.
> > > * Thank you for the great point. As the reviewer pointed out, the methods such as Data Shapley or TracIn that use additional validation/test set to directly measure the impact of individual samples on the test accuracy can more easily distinguish mislabeled samples from other correctly labeled samples. On the other hand, our CG-score as well as other methods including C-score, EL2N, and forgetting score, which does not assume any validation dataset, determines the value of each instance based on the irregularity of each training sample with respect to the data distribution, which often makes the mislabeled samples have higher scores. We will highlight this difference and explain it in the final version.
> > > * Thank you for the suggestion. We will add comments on the required calculation complexity for ImageNet. As we explained in Appendix A, we have determined the required number of subsampling rounds based on the size of the dataset and the number of classes. The subsampled size is usually limited by the maximum size of the matrix that we can take the inversion. For the ImageNet dataset, where there are 1,000 classes and about 1,300 images per each class, when we use the subsampling ratio of 1:19 (thus making the subsampled matrix size as 26,000*26,000), the required number of stochastic runs to cover a half of the samples from 999 classes will be approximately 25. Since there are 1,000 classes, this means that we need 25,000 stochastic runs. This is a huge computation, but still can be more efficient than calculating Data Shapley or C-score, which requires re-training of a model with the number of times exponentially/linearly proportional to the sample size.

---

> ### Author Response · Authors · 2022-11-18
> **Response to Reviewer egMu (2/3)**
>
> >**2. Add an appendix section explaining the derivations of eq. 3-4 using the Shur complement.**
>
> We added more details on the derivations of eq. 3-4 in Appendix K.
>
>
> >**3. Could the authors provide more information about the running-time they observe in practice? How does the large matrix inversion $(\mathbf{H}^\infty)^{-1}$ compare to a single training run, or how much the computation is reduced when performing the stochastic calculation?**
>
> In Table 4 of Appendix C.2, we compared the computation time (in seconds) of our CG-score with other methods. The computation time of our method depends on the dimension of the matrix we take the inversion. For CIFAR-100 dataset, each class includes 500 instances and when we calculate the CG-score by 1:4 sampling ratio between the class of interest and the rest of the classes, we need to calculate the inversion of 2,500 $\times$ 2,500 matrices. This calculation can be done within less than 1 minute, which is 1/10x  compared to the training time of ResNet18 for 20 epochs to get the EL2N score. The forgetting score, which requires the full training until 200 epochs, requires 100x times compared to the computation time of our method. For CIFAR-10 dataset, on the other hand, where each class includes 5,000 instances, with the same 1:4 sampling ratio, we need to take the inversion of 25,000 $\times$ 25,000 matrix, which requires 1.2 hours to calculate the score. This time is comparable to the training time of ResNet18 for 200 epochs (the time to calculate the Forgetting score). All the baselines as well as our method require stochastic calculation, averaging the score over multiple runs. For a fair comparison, we compared the time to get each score for a single run.
>
> >**4. In the experiments, the C-score doesn't appear to have been used for Fashion-MNIST (Figure 2). Is there a reason for this, would the authors be able to add it?**
>
> In [a], C-score was calculated for CIFAR-10 and CIFAR-100 but not for FMNIST. Since calculating C-score requires re-training of a neural network at least the number of repetitions proportional to the size of the dataset, 60,000 for FMNIST, we did not additionally calculate the score for this comparison.
>
> [a] Jiang et al., Characterizing Structural Regularities of Labeled Data in Overparameterized Models, ICML 2021.
>
> >**5. It would be nice to allude to several implementation details in the main text.**
>
> Thank you for the suggestion. We added sentences in the main text to point where the implementation details can be found in the appendix.

---

> ### Author Response · Authors · 2022-11-18
> **Response to Reviewer egMu (1/3)**
>
> We sincerely thank the reviewer for the constructive feedback. We tried our best to address the reviewer’s concerns and questions. Hope this response answers the reviewer’s questions.
>
> >**1. One shortcoming in this approach seems to be that we expect large complexity gap scores both for useful/difficult samples and for samples with incorrect/noisy labels. I suspect this issue does not exist for any previous method that explicitly evaluates performance on a validation/test set when training with partial training sets. Could this be discussed/acknowledged somewhere in the paper?**
>
> Discriminating mislabeled data and atypical (but useful) data is a major challenge in data valuation, since both the mislabeled data and atypical data are irregular in the data distribution. Similar challenges have been observed in other scores based on training dynamics, including EL2N and forgetting score. However, we found that mislabeled data usually has a higher CG-score than atypical data, and this tendency gives us the possibility to separate mislabeled data from the rest of the clean data.
>
> To check the ability of CG-score in separating mislabeled examples from the atypical examples,  we additionally performed a data window experiment. The data instances are sorted in ascending order by the CG-score, and we compared the test accuracy of a neural network trained with 50\% of training instances selected from offset\% to (offset+50)\% scoring group, for different offset points of $\{0,5,10,\dots, 45,50\}$. For example, when the offset is 20\%, we select the data instances from 20\% to 70\% scoring examples. When the training instances do not include mislabeled data, we can expect that the window experiment will show higher accuracy as the offset increases up to 50\%. We added 20\% of random label noise to FMNIST and CIFAR-100 datasets to see how the trend changes when the dataset includes mislabeled instances.
>
> Table. Window experiment with varying offset(\%). Samples are sorted in ascending order.
> | Offset(%) | Random(baseline) |   0   |   5   |  10   |  15   |  20   |  25   |  30   |  35   |  40   |  45   |  50   |
> |:---------:|:----------------:|:-----:|:-----:|:-----:|:-----:|:-----:|:-----:|:-----:|:-----:|:-----:|:-----:|:-----:|
> |  FMNIST   |      86.78       | 89.50 | 90.28 | 90.84 | 91.30 | 91.80 | 91.90 | 91.11 | 87.56 | 74.2  | 55.50 | 44.50 |
> |  CIFAR10  |      81.87       | 84.19 | 85.14 | 85.45 | 85.80 | 85.17 | 84.54 | 83.04 | 80.33 | 74.90 | 67.84 | 58.41 |
>
>
> As shown in Fig. 6 of Appendix B (summarized as table above), for FMNIST, the test accuracy increases until the offset reaches 30\% and then drops after the point. When the offset is 30\%, the 50\%-width window includes 30\% to 80\% scoring examples. Since 20\% mislabeled data are mainly located within the 80\% to 100\% scoring group, as the offset increases above 30\%, the 50\%-width window starts to include mislabeled instances and this causes the rapid drop of the test accuracy. Thus, from the window experiments we can see that the mislabeled data has the highest CG-score and can be separable from the rest of the clean examples by the CG-score.
>
> Then, the next reasonable question is how to set the threshold on CG-score to detect the mislabeled data when the portion of mislabeled data is unknown. We argue that the sign of the partial CG-score $y_i (\mathbf{y}_{-i}^\top \mathbf{h}_i)$, which is a sub-term of the CG-score including the label information $y_i$, can be used in this purpose. As explained in Sec. 2.3, the partial CG-score measures the gap between the average similarity of the data instance from the same-class examples to the different class examples. Thus, by checking the sign of the partial CG-score, we can discriminate mislabeled samples from clean samples. In Fig. 6, we show the scatter plot of clean (blue) and mislabeled data (orange), where one can find that mislabeled examples tend to have positive partial CG-score. Furthermore, as shown in Table 2, we can check that for FMNIST with 20\% label noise (12,000 mislabeled instances and 48,000 clean instances), 98\%(=11796/12000) of mislabeled data ends up having positive partial CG-score, while only 7\%(=3462/44538) of clean data has positive partial CG-score. Thus, even when the portion of label noise is unknown, our partial CG-score can effectively detect the mislabeled data by the sign information. This tendency was less clear for CIFAR-10 dataset due to the increased dataset complexity, but still the tendency existed.

---

### Official Review · Reviewer_QwGQ · 2022-10-24

**Confidence:** 4
**Correctness:** 3
**Technical Novelty And Significance:** 3
**Empirical Novelty And Significance:** 3
**Recommendation:** 6

**Clarity, Quality, Novelty And Reproducibility:**

The work is clearly written and organized and high-quality in terms of sound development and analyses of the approach (aside from perhaps not situating it with respect to some related work).  As mentioned, although the basic approach of measuring the removal of an item to measure its importance or impact is a common one in ML - this is a novel application of it on this bound and an interesting idea, and there is significant development beyond this starting point.

The authors provided their code in supplementary material as well - so reproducibility should be good.

**Strength And Weaknesses:**


Strengths:
1) The approach is interesting (and thought-provoking) and soundly developed.  It is not purely a trivial application of the prior developed error bound as a lot of thought went into how to apply it and efficiently compute it, and a lot of analyses to understand what is doing and measuring, and how it relates to other metrics.

2) Compared to recent past work analyzing data importance from neural net training it has clear advantage in not requiring training any neural network.

3) Thorough and varied experiments are provided to illustrate the properties of the metric.

4) The paper is well written and organized.


Weaknesses:

1) I feel perhaps the biggest weakness is lack of mention and comparison to prior work targeting the particular problems this metric is proposed to be applied to - selecting a subset of data for more efficient training, and finding mislabeled training data.

In particular - a major motivation of the approach was to "dataset pruning" - which is essentially using the score per data instance to select a subset of data points for training, e.g., in order to enable fast / less expensive training on just a subset of the data.  The paper is a bit misleading in stating it is the only approach to doing this that doesn't require training a neural network first - as in general this is an area of research with a long history, and there are many methods that have been developed to select a subset of the data for training that don't all require fitting a particular predictive model to it first.  It would be best to mention these methods, why this proposed approach may have some advantage, if any, and also include some of these in the experiment results.  In particular - would I use this proposed metric, or one existing approach to identifying a subset of the data to use for training?  What are the advantages and disadvantages of each?  Additionally there are many other alternatives to accelerating machine learning training such as data sketches - not just selecting a subset of data, but summarizing the data in some way first (perhaps even with random features) to speedup training using a smaller transformed set of data.

Some examples of papers in this area:
- "Coresets for Data-efficient Training of Machine Learning Models" Mirzasoleiman et al.
- "Subset Selection in Machine Learning: Theory, Applications, and Hands On" (tutorial) Iyer et al.
- "Training Data Subset Selection for Regression With Controlled Generalization Error" Sivasubramanian et al.
- "Introduction to Core-sets: an Updated Survey" Feldman
- "GLISTER: Generalization based Data Subset Selection for Efficient and Robust Learning" Killamsetty et al.
- "Finding High-Value Training Data Subset through Differentiable Convex Programming" Das et al.
- "Oblivious Sketching for Logistic Regression" Munteanu et al.

The same also goes for handling label error or detecting which data instances are mislabeled - again there is a large body of related work in this area not mentioned or compared to.  E.g.:
- "An instance level analysis of data complexity" Smith et al.
- "Identifying mislabeled training data." Brodley et al.

I found myself wondering throughout, how this work sits compared to these other approaches.  Overall it may be this is more of an analyses of a particular metric derived from a bound - so may not be claiming to be better than other approaches for each problem, but in order to have a complete and thoroughly useful analysis, it would be helpful to know how it compares.

2) The original bound and the corresponding proposed metric is based on a single hidden layer MLP relu neural net - with a lot of further restricting assumptions such as on the data and weights, and also with only the hidden layer parameters being trained in the process.  The analyses of the proposed metric also make various other assumptions, of varying strength.  It would be helpful to see more experiments and analyses to understand how the proposed approach performance changes as these assumptions are violated - especially since the datasets and variations used with them were limited (just 3 datasets with no exploration of varying model types, for example).

It would also be helpful for some discussion around what it would mean for the metric with changing neural net architecture or assumptions - how would it diverge, what would a good metric change in these cases, etc.

I.e., what about different kinds of nets, e.g., more complicated (like transformer / self-attention), or with different form (e.g., different types of convolution or RNN) or activation, or with more layers?  The intrinsic biases the different network configurations and architectures correspond encode would seem to suggest different metrics might be better suited to different architectures - so what kind of universality does the proposed metric have?

3) The two objectives / proposed uses for the proposed metric seem at odds and hard to reconcile. The proposed metric simultaneously is higher if a training example is necessary for good generalization (e.g., close to the decision boundary), but also higher if the training example is mislabeled.  It's being used for both things but both would require the user to take opposite actions - if the data instance has a high score and if we assume it is an important training instance we want to keep it in the training data over other instances.  On the other hand if the data instance has a high score, and we assume it is a mislabeled data instance, we would want to remove it from the training data.  How then could one decide based on this metric whether or not exclude a data instance?  I.e., how can you differentiate such cases?


4) The idea is pretty straightforward - in the sense of taking an error bound and just measuring the difference on it by removing a data instance - this kind of leave-out-out analyses approach has been widely used and is common in machine learning (e.g., for feature importance as well), in many different context.  This may be seen as somewhat limiting the novelty, but I feel the additional development of a computational efficient algorithm and analyses of the approach overcomes this limitation.



**Summary Of The Paper:**

The authors propose a new data instance valuation metric (CG score) - for scoring the how impactful a particular training instance would be on training a neural net.  This can be viewed in terms of how difficult of an example it is to learn and how much it effects generalization, etc.  This metric does not require actually training any neural net (for any number of steps), unlike much previous work on the topic.  The approach is based on a generalization error bound for 1-hidden-layer neural networks (under a number of simplifying assumptions) - in particular, measuring the change in this bound if a particular data instance is excluded from the training set.  They develop a succinct and computational efficient method to compute this change for each data instance.

They propose using such a metric for data pruning before training - to reduce the size of the data set used / needed for training a network, and for identifying mislabeled data points - and validate this with analyses and experiments.  They also demonstrate it could identify those data instances that are harder to learn - which suggests it could also be useful for something like curriculum learning.


**Summary Of The Review:**

Overall I found it to be an interesting and potentially useful paper on the one hand that could inspire further research, but on the other hand also raising several questions and missing fields of prior work on the target applications (i.e., dataset subset selection / pruning, and handling label error).  In particular I feel it is not really accurate to claim past methods have not been developed to quantify data instance complexity without also training the predictive model, and that the missing related work and describing where this work fits in comparison could have a negative effect, as related work may end up being rehashed if readers are unaware of it.  Further it is not enough to determine if the proposed approach should be used in practice (i.e., the practical benefit) vs. the other methods not mentioned or compared to.  I hope these concerns can be addressed in revisions and in appendices as needed as well.

---

> ### Author Response · Authors · 2022-11-18
> **Response to Reviewer QwGQ (3/3)**
>
> >**3. The two objectives / proposed uses for the proposed metric, 1) detecting training examples necessary for generalization and 2) detecting mislabeled training examples, seem at odds and hard to reconcile. How then could one decide based on this metric whether or not to exclude a data instance? i.e., how can you differentiate such cases?**
>
> Discriminating mislabeled data and atypical (but useful) data is a major challenge in data valuation, since both the mislabeled data and atypical data are irregular in the data distribution. We found that mislabeled data usually has a higher CG-score than atypical data, and this tendency gives us the possibility to separate mislabeled data from the rest of the clean data.
>
> To check the ability of CG-score in separating mislabeled examples from the atypical examples,  we additionally performed a data window experiment. The data instances are sorted in ascending order by the CG-score, and we compared the test accuracy of a neural network trained with 50\% of training instances selected from offset\% to (offset+50)\% scoring group, for different offset points of $\\{0,5,10,\dots, 45,50\\}$. For example, when the offset is 20\%, we select the data instances from 20\% to 70\% scoring examples. When the training instances do not include mislabeled data, we can expect that the window experiment will show higher accuracy as the offset increases up to 50\%. We added 20\% of random label noise to FMNIST and CIFAR-100 datasets to see how the trend changes when the dataset includes mislabeled instances.
>
> Table. Window experiment with varying offset(\%). Samples are sorted in ascending order.
> | Offset(%) | Random(baseline) |   0   |   5   |  10   |  15   |  20   |  25   |  30   |  35   |  40   |  45   |  50   |
> |:---------:|:----------------:|:-----:|:-----:|:-----:|:-----:|:-----:|:-----:|:-----:|:-----:|:-----:|:-----:|:-----:|
> |  FMNIST   |      86.78       | 89.50 | 90.28 | 90.84 | 91.30 | 91.80 | 91.90 | 91.11 | 87.56 | 74.2  | 55.50 | 44.50 |
> |  CIFAR10  |      81.87       | 84.19 | 85.14 | 85.45 | 85.80 | 85.17 | 84.54 | 83.04 | 80.33 | 74.90 | 67.84 | 58.41 |
>
>
> As shown in Fig. 6 of Appendix B (summarized as table above), for FMNIST, the test accuracy increases until the offset reaches 30\% and then drops after the point. When the offset is 30\%, the 50\%-width window includes 30\% to 80\% scoring examples. Since 20\% mislabeled data are mainly located within the 80\% to 100\% scoring group, as the offset increases above 30\%, the 50\%-width window starts to include mislabeled instances and this causes the rapid drop of the test accuracy. Thus, from the window experiments we can see that the mislabeled data has the highest CG-score and can be separable from the rest of the clean examples by the CG-score.
>
> Then, the next reasonable question is how to set the threshold on CG-score to detect the mislabeled data when the portion of mislabeled data is unknown. We argue that the sign of the partial CG-score $y_i (\mathbf{y}_{-i}^\top \mathbf{h}_i)$, which is a sub-term of the CG-score including the label information $y_i$, can be used in this purpose. As explained in Sec. 2.3, the partial CG-score measures the gap between the average similarity of the data instance from the same-class examples to the different class examples. Thus, by checking the sign of the partial CG-score, we can discriminate mislabeled samples from clean samples. In Fig. 6, we show the scatter plot of clean (blue) and mislabeled data (orange), where one can find that mislabeled examples tend to have positive partial CG-score. Furthermore, as shown in Table 2, we can check that for FMNIST with 20\% label noise (12,000 mislabeled instances and 48,000 clean instances), 98\%(=11796/12000) of mislabeled data ends up having positive partial CG-score, while only 7\%(=3462/44538) of clean data has positive partial CG-score. Thus, even when the portion of label noise is unknown, our partial CG-score can effectively detect the mislabeled data by the sign information. This tendency was less clear for CIFAR-10 dataset due to the increased dataset complexity, but still the tendency existed.

---

> ### Author Response · Authors · 2022-11-18
> **Response to Reviewer QwGQ (2/3)**
>
> >**2. It would also be helpful for some discussion around what it would mean for the metric with changing neural net architecture or assumptions. What kind of universality does the proposed metric have? More experiments and analyses to understand how the proposed approach performance changes as the assumptions on the analysis are violated or the neural network architectures change.**
>
> The CG-score was designed by using the gap in the generalization error bounds of the overparameterized ReLU activated two-layer network, $\mathbf{y}^\top (\mathbf{H}^\infty)^{-1} \mathbf{y}$  where the Gram matrix $\mathbf{H}^\infty$ captures the training dynamics of the considered two-layer neural network. The Gram matrix is equivalent to the expectation for Neural Tangent Kernel (NTK) [a] of the two-layer infinite-width MLP at the initialization.
>
> When we consider a more general neural network model, the only part that needs to be adjusted in the definition of the CG-score is the NTK part, $\mathbf{H}^\infty$. More specifically, we need to replace $\mathbf{H}^\infty$ by the NTK of the considering neural network model,
> $
> \mathbf{H}_{ij}(\mathbf{\theta})=\nabla_\theta f(\mathbf{x}_i, \mathbf{\theta})^\top \nabla_\theta f(\mathbf{x}_j, \mathbf{\theta})
> $
> where $f(\mathbf{x}, \mathbf{\theta})$ is the output of the network with parameter $\mathbf{\theta}$ at input $\mathbf{x}$. For complicated NNs, however, the computation of this type of Gram matrix requires high-computational cost, especially when the dimension of $\mathbf{\theta}$, the parameter space of the network, is high. For the two-layer NNs considered in our paper, this heavy calculation has been avoided, since the NTK converges to the Gram matrix $\mathbf{H}^\infty$ that only depends on the data instances.
>
> The remaining question is then how well the CG-score, calculated based on the NTK of the simple two-layer MLP, performs in more general neural networks. We showed the applicability of the proposed CG-score in more complicated/general models by using diverse empirical observations. In our original paper, we evaluated the CG-score for data pruning experiments, in ResNet18/50 (Fig.2), to show the robustness of our score against model change. We also additionally conducted evaluations of the CG-score in DenseNet (Fig.9), which is a relatively more complex architecture than ResNet, in the revised manuscript. Our score achieves competitive performances compared to other baselines (EL2N or forgetting scores) which are evaluated on the network itself where the test accuracy is measured.
>
> Here, we did not evaluate our scores in more complicated networks such as Vision Transformer (ViT), one of the reviewer’s suggested models, since ViT uses a pre-trained model, trained on large amount of data (ImageNet, ImageNet-21k, JFT-300M), and then is fine-tuned to the target domain. Since our data valuation methods, including ours, do not evaluate the value of each sample from the perspective of fine-tuning, ViT might not be a suitable network to compare the performances of data valuation. ResNet or DenseNet, on the other hand, are trained using the selected data instances from random initialization, which makes them more suitable architectures to evaluate different valuation methods.
>
>
> [e] Jacot et al., Neural Tangent Kernel: Convergence and Generalization in Neural Networks, NIPS 2018.

---

> ### Author Response · Authors · 2022-11-18
> **Response to Reviewer QwGQ (1/3)**
>
> We sincerely thank the reviewer for the constructive feedback. We tried our best to address the reviewer’s questions.
>
> >**1. Comparison to prior work targeting the particular problems this metric is proposed to be applied to - selecting a subset of data for more efficient training, and finding mislabeled training data**
>
> We appreciate the reviewer for the insightful feedback. As the reviewer suggested, we added comparisons of our work to more prior works, targeting each of the applications 1) selecting a subset of data for more efficient training and 2) finding mislabeled training data in Appendix D. For 1), we included CRAIG [a] and TracIn [b] in the experiment, and for 2), TracIn [b].
>
> As the reviewer pointed out, coresets can be another representative method in selecting a subset of data for more efficient training. In particular, CRAIG [a] provides generic algorithms to select a weighted representative subset that closely approximate the full gradient and achieves the one of the best performances in training NNs among coreset selection schemes. Thus, we used CRAIG as a representative baseline for coresets. However, we'd like to emphasize that CRAIG requires the training of a model to find the coresets that approximate the full gradient. Moreover, CRAIG keeps updating the coresets over the training, and thus it may not be a suitable algorithm for dataset pruning applications, where we want to select a “fixed” subset of data (possibly at the beginning of the training) that will be used throughout the training. Nonetheless, we compared the performance of CRAIG in the dataset pruning experiment with our method in Figure 7-(a) of Appendix D.2. The corsets selected from CRAIG are updated every 10 epochs, a total of 20 times for 200 epochs of training.  With the updated coresets, CRAIG indeed could achieve a better performance than our method in the pruning experiment. This result might imply the effectiveness of the scheduled batch selection, e.g., curriculum learning, in training neural networks.
>
> Another interesting baseline the reviewer mentioned is the dataset condensation/distillation, where the task is to synthesize a small dataset such that the model trained on it achieves high performance on the original large dataset. However, we would like to highlight that the “scale of synthesized dataset” considered in dataset condensation is much smaller than the scale of data selection, i.e., for most dataset condensation methods, the images per class (IPC) range from 1, 10 to 50. For CIFAR-10, where there are 5,000 images per class, IPC of 50 is equal to the pruning ratio of  99\%. In this extreme range, we believe that dataset condensation will outperform most of the data selection methods. For CIFAR-100, on the other hand, IPC of 50 is equal to the pruning ratio of 90\%, less severe than that of CIFAR-10. One surprising observation from previous work [d] is that for CIFAR-100 with the pruning ratio of 90\% (IPC 50), simple data selection method, e.g., K-center, often outperforms the best performing condensation methods [d]. This result shows that there will be some sweet spot on IPC where the winner between the dataset condensation and data selection might change. We think this is an interesting open problem.
>
> As the reviewer suggested, we also compared the performance of our CG-score in detecting mislabeled data with another baseline targeting the noise label detection, TracIn [b], which identifies mislabelled data by identifying 'self-influence' of each training example, i.e., the influence of a training point on its own loss during the training process whenever the training example of interest was utilized. In [b], it was shown that mislabelled examples have highest TracIn values, and TracIn values can be effectively used in identifying mislabelled examples. In Fig. 8 of Appendix D, we show the comparison of our method with TracIn in identifying mislabeled examples in FMNIST and CIFAR-100 datasets including 20\% label noise. TracIn achieved better performance in CIFAR-100, but ours outperformed TracIn in FMNIST. Since TraIn measures the 'self-influence' of each training example over the training, starting from the 20th epoch, for relatively simpler dataset such as FMNIST, some mislabeled instances could have already been memorized at the 20th epoch, which makes them not detectable by the TracIn values. On the other hand, our method better detects mislabelled dataset for easier datasets as discussed in Sec. 3.2. Thus, we can conclude that depending on data complexity, the outperforming method can be changing.
>
>
> [a] Mirzasoleiman et al., Coresets for Data-efficient Training of Machine Learning Models, ICML 2020.
>
> [b] Pruthi et al., Estimating Training Data Influence by Tracing Gradient Descent, NeurIPS 2020.
>
> [c] Killamsetty et al., GLISTER: Generalization based Data Subset Selection for Efficient and Robust Learning, AAAI 2021.
>
> [d] Cui et al., DC-BENCH: Dataset Condensation Benchmark, NeurIPS 2022.

---

> ### Author Response · Authors · 2022-12-07
> **A kind reminder for Reviewer QwGQ**
>
> Dear reviewer QwGQ,
>
> Thank you for your effort in reviewing our work.
>
> We would like to kindly remind you to check our author's response if you have not already. We would appreciate it if you could inform us whether our response successfully addressed your concerns. Even a short statement (why our response did or didn't change your assessment of our work) would be of great help to us. Thank you!

---

### Official Review · Reviewer_6ZH6 · 2022-10-25

**Confidence:** 4
**Correctness:** 3
**Technical Novelty And Significance:** 2
**Empirical Novelty And Significance:** 2
**Recommendation:** 6

**Clarity, Quality, Novelty And Reproducibility:**

- The paper itself is well written and easy to follow.
- The proposed method is somewhat heavily based on the previous work (Arora et al., 2019) but applied to the new problem. Thus, the novelty is somewhat marginal.

**Details Of Ethics Concerns:**

Not applicable.

**Strength And Weaknesses:**

Strength:
- The proposed method is computationally much more efficient than alternatives.
- Still, the proposed method shows competitive empirical results in comparison to alternatives.
- The proposed method is well supported with the theoretical analyses.

Weakness:
- It is unclear whether the data value can be "independent" of the model.
- In some experimental results, the proposed method shows somewhat consistently worse performance.
- It would be good if the authors can provide empirical advantages of computational complexity.

**Summary Of The Paper:**

- The authors tried to tackle a fundamental problem, data valuation.
- Most previous works need model training for quantifying the values of the samples (which is computationally inefficient); however, the proposed method does not need model training.
- The authors provided various experimental results that show the superiority and usefulness of the proposed data valuation in multiple datasets (including CIFAR-10)

**Summary Of The Review:**

1. Generalization
- In this paper, the proposed method is highly dependent on two-layer overparameterized MLP. (H in Equation (2))
- In that case, can we say that this score is generalized to more general neural networks? Or, can we say this score is generalized to simpler models such as linear models?
- Can we also generalize this method for a regression problem?

2. Connection between data value and model
- The value of the data can be different across different models.
- For instance, the value of the sample with a linear model can be different with a non-linear model.
- In that case, how can the proposed method quantify the value of the data depending on the models?
- The author said that this model is independent of the model but I am not sure in general whether data value can be independent of the model.

3. Experiments
- The experimental results are interesting.
- Especially, the authors show consistent improvements for noisy label finding.
- It would be interesting if the authors can provide the results with a higher noisy ratio (like even higher than 50%) to check the robustness of the model.

4. Figure 2
- For pruning lower value samples, the proposed method shows somewhat consistently worse performance than alternatives.
- Is there any intuition that the proposed method does not do well for discovering lower value samples? Especially with a smaller training data portion?

---

> ### Author Response · Authors · 2022-11-18
> **Response to Reviewer 6ZH6 (2/2)**
>
> >**2. Empirical advantages of computational complexity.**
>
> In Table 4 of Appendix C.2, we compared the computation time (in seconds) of our CG-score with other methods. The computation time of our method depends on the dimension of the matrix we take the inversion. For CIFAR-100 dataset, each class includes 500 instances and when we calculate the CG-score by 1:4 sampling ratio between the class of interest and the rest of the classes, we need to calculate the inversion of 2,500 $\times$ 2,500 matrices. This calculation can be done within less than 1 minute, which is 1/10x  compared to the training time of ResNet18 for 20 epochs to get the EL2N score. The forgetting score, which requires the full training until 200 epochs, requires 100x times compared to the computation time of our method. For CIFAR-10 dataset, on the other hand, where each class includes 5,000 instances, with the same 1:4 sampling ratio, we need to take the inversion of 25,000 $\times$ 25,000 matrix, which requires 1.2 hours to calculate the score. This time is comparable to the training time of ResNet18 for 200 epochs (the time to calculate the Forgetting score).
>
> >**3. It would be interesting if the authors can provide the results with a higher noisy ratio (like even higher than 50%) to check the robustness of the model.**
>
> As the reviewer suggested, we conducted additional experiments to check the detectability of mislabeled instances by our CG-score, when the noise ratio increases to 50\% and even to 80\% for FMNIST and CIFAR-10 dataset. The results are shown in Fig. 14 of Appendix J.  In the CIFAR-10 dataset, when the noise ratio is 80\%, each class includes 1,000 correctly labeled images and 4,000 mislabeled images, composed of 445 samples coming from each of the other nine classes. Even for such an extremely noisy case, our CG-score could effectively detect the mislabeled data, since our score can discover samples that have a relatively lower correlation to the majority of the samples of  the same class as explained in Sec. 2.3.
>
> >**4. Is there any intuition that the proposed method does not do well for discovering lower value samples? Especially with a smaller training data portion?**
>
> We can explain this tendency by the fact that the distribution of CG-score is highly concentrated around low values with a peak (delta function) around value 0.  In Fig. 13 of Appendix H, we show the density of the CG-score for three datasets FMNIST, CIFAR-10/100. Since low-scoring samples are highly concentrated around value 0, it could be hard to distinguish the lowest-valued samples. On the other hand, we can observe a long-tail for the CG-score histograms of CIFAR-10/100 dataset. It means that the high-scoring samples are in fact distinguishable from each other by the CG-score values. Thus, we expect that CG-score could better discover high value samples than low value samples.
>
> The reason that we observe a rapid performance degradation at a smaller training data portion in Fig. 2 (a) is due to the leave-one-out method we used in the calculation of the CG-score. Removing a typical sample from a dataset does not change the generalization error bounds much, since similar samples already exist in the dataset. Thus, typical samples tend to have low CG-scores when the score is measured by the leave-one-out method. However, when we remove 50\% of instances, samples sharing the typicality can be excluded simultaneously from the dataset, which might cause severe degradation of the generalization capability of the neural network. Thus, for a smaller training data portion, it can be better to make sure at least a small portion of typical samples is indeed included in the training. Similar observations have been made in [c].
>
> [c] Swayamdipta et al., Data Cartography: Mapping and Diagnosing Datasets with Training Dynamics, EMNLP 2020.

---

> > ### Comment · Reviewer_6ZH6 · 2022-11-28
> > **Thank you for the authors' thoughtful answers to my comments.**
> >
> > In general, the rebuttal resolved most of my previous comments (including intuitions of worse performance with smaller training data).
> > Also, thank you for providing additional results including computational complexity analyses and higher noise ratio.
> >
> > However, it is still unclear whether this score can be generalized to any model. (Like a simple linear model).
> > Also, it is good to clearly mention the limitations of this work (including discovering lower value samples and smaller sample regime) in the camera-ready version (if accepted).
> >
> > I think this is a good paper but not the level of "clear accept (8)".
> > Thus, I stand on my original score (6) after reading the rebuttals and other reviewers' comments.

---

> > > ### Author Response · Authors · 2022-12-02
> > > **Thank Reviewer 6ZH6 for the reply.**
> > >
> > > We are glad to hear that our rebuttal resolved most of your previous concerns. Thank you for the suggestion to better explain the limitations of work. We will incorporate your feedback in the final version.

---

> ### Author Response · Authors · 2022-11-18
> **Response to Reviewer 6ZH6 (1/2)**
>
> We sincerely thank the reviewer for the constructive feedback. We tried our best to address the reviewer’s concerns and questions. Hope this response answers the reviewer’s questions.
>
> >**1. Can the data valuation be independent of the model? Can the CG-score, proposed based on the analysis of the two-layer overparameterized MLP, be generalized to more general neural networks/simpler models/regression models?**
>
> The CG-score was designed by using the gap in the generalization error bounds of the overparameterized ReLU activated two-layer network, $\mathbf{y}^\top (\mathbf{H}^\infty)^{-1} \mathbf{y}$  where the Gram matrix $\mathbf{H}^\infty$ captures the training dynamics of the considered two-layer neural network. The Gram matrix is equivalent to the expectation for Neural Tangent Kernel (NTK) [a] of the two-layer infinite-width MLP at the initialization.
>
> When we consider a more general neural network model, the only part that needs to be adjusted in the definition of the CG-score is the NTK part, $\mathbf{H}^\infty$. More specifically, we need to replace $\mathbf{H}^\infty$ by the NTK of the considering neural network model,
> $
> \mathbf{H}_{ij}(\mathbf{\theta})=\nabla_\theta f(\mathbf{x}_i, \mathbf{\theta})^\top \nabla_\theta f(\mathbf{x}_j, \mathbf{\theta})
> $
> where $f(\mathbf{x}, \mathbf{\theta})$ is the output of the network with parameter $\mathbf{\theta}$ at input $\mathbf{x}$. For complicated NNs, however, the computation of this type of Gram matrix requires high-computational cost, especially when the dimension of $\mathbf{\theta}$, the parameter space of the network, is high. For the two-layer NNs considered in our paper, this heavy calculation has been avoided, since the NTK converges to the Gram matrix $\mathbf{H}^\infty$ that only depends on the data instances.
>
> Then, the remaining question is how well the CG-score, calculated based on the NTK of the simple two-layer MLP, performs in more general neural networks. We argue the applicability of the proposed CG-score in more complicated/general models by using two empirical observations. First, in our paper, we evaluated the CG-score for data pruning experiments, in ResNet18/50 (Fig.2) and in DenseNet (Fig.9) to demonstrate the effectiveness of our score in complicated neural networks. Our score achieves competitive performances compared to other baselines (EL2N or forgetting scores) which are evaluated on the network itself where the test accuracy is measured. Second, some recent work [b] has empirically demonstrated that the generalization error bound $\mathbf{y}^\top (\mathbf{H})^{-1} \mathbf{y}$ with the NTK matrix $\mathbf{H}$ is robust against model changes. The robustness of $\mathbf{y}^\top (\mathbf{H})^{-1} \mathbf{y}$ implies that the eigenspace and eigenspectrum of $\mathbf{H}$, defined by the eigenvectors/eigenvalues of $\mathbf{H}$, maintains a close correlation with the label vector $\mathbf{y}$. In fact, this tendency needs to be held for a neural network to successfully classify the training instances based on their labels. Since our CG-score is defined in terms of the gap between the generalization error bounds, we can infer the robustness of our CG-score against model changes from the robustness of the generalization error bounds.
>
> [a] Jacot et al., Neural Tangent Kernel: Convergence and Generalization in Neural Networks, NIPS 2018.
>
> [b] Wu et al., DAVINZ: Data Valuation using Deep Neural Networks at Initialization, ICML 2022.

---

### Author Response · Authors · 2022-11-18
**The revised paper is uploaded**

We sincerely thank the reviewers for the constructive feedback. We revised the main document and appendix of our paper to include additional materials that address the reviewers’ questions.  We have uploaded our revised paper with the following main modifications:

* (Appendix A.3) We explain details on stochastic method to calculate CG-scores.
* (Appendix B) We show the effectiveness of CG-score in separating mislabeled examples from atypical examples with additional experiments.
* (Appendix C.2) We report the averaged computation times of CG-score and other baselines.
* (Appendix D and E) We add results on data pruning experiments with 1) additional baselines, TracIn[a] and CRAIG[b] (Appendix D), and 2) additional network, DenseNet (Appendix E).
* (Appendix D and H) We add results on noise detection experiments with 1) additional baseline, TracIn[a] (Appendix D), and 2) at higher noise rate (Appendix H).
* (Appendix G) We analyze Tiny ImageNet, which is more complicated dataset than CIFAR-10/100, with CG-score.
* (Appendix J) We add analysis on feature-space CG-score.
* (Appendix K) We add detailed explanation of Schur complement.

The main revision materials are highlighted in blue.

[a] Pruthi et al., Estimating Training Data Influence by Tracing Gradient Descent, NeurIPS 2020.

[b] Mirzasoleiman et al., Coresets for Data-efficient Training of Machine Learning Models, ICML 2020.

---

### Decision · Program_Chairs · 2023-01-20

**Decision:**

Accept: poster

**Justification For Why Not Higher Score:**

Lack of experiments or discussion for more complex datasets.

**Justification For Why Not Lower Score:**

The paper is overall strong-- it presents some nice and useful contributions which are both empirically and theoretically justified.

**Metareview: Summary, Strengths And Weaknesses:**

The paper proposes a data valuation score that can effectively be used as a proxy for the impact of individual datapoints in the optimization/generlization of DNNs called the complexity-gap score. This score finds irregular or mislabeled data instances and can be computed without needing to train the model.

The proposed method tackles an important and fundamental problem in ML regarding data valuation and data quality. The method is more efficient than the alternatives and is shown to be effective on a range of tasks. Moreover, the idea is proposal and nicely justified both empirically and theoretically.

The main weakness is that it's unclear whether results hold on more complex data-- in such settings, the intermediate representations of the model may be less reliable. The authors are encouraged to address these possible limitations.

**Note From Pc:**

if the above contains the word "oral" or "spotlight" please see: "oral" presentation means -> notable-top-5% and "spotlight" means -> notable-top-25%. As stated in our emails, we are disassociating presentation type from AC recommendations